# Neural representation of visual concepts in people born blind

Ella Striem-Amit [1,2], Xiaoying Wang[3], Yanchao Bi[3] & Alfonso Caramazza[1,4]

How do we represent information without sensory features? How are abstract concepts like "freedom", devoid of external perceptible referents, represented in the brain? Here, to address the role of sensory information in the neural representation of concepts, we used fMRI to investigate how people born blind process concepts whose referents are imperceptible to them because of their visual nature ("rainbow", "red"). Activity for these concepts was compared to that of sensorially-perceptible referents ("rain"), classical abstract concepts ("justice") and concrete concepts ("cup"), providing a gradient between fully concrete and fully abstract concepts in the blind. We find that anterior temporal lobe (ATL) responses track concept perceptibility and objecthood: preference for imperceptible object concepts was found in dorsal ATL, for abstract (non-object, non-referential) concepts in lateral ATL, and for perceptible concepts in medial ATL. These findings point to a new division-of-labor among aspects of ATL in representing conceptual properties that are abstract in different ways.

[1] Department of Psychology, Harvard University, Cambridge, MA 02138, USA. [2] Department of Psychology, Ben Gurion University of the Negev, Beer-Sheva 8410501, Israel. [3] State Key Laboratory of Cognitive Neuroscience and Learning & IDG/McGovern Institute for Brain Research, Beijing Normal University, Beijing 100875, China. [4] Center for Mind/Brain Sciences, University of Trento, 38068 Rovereto, Italy. These authors contributed equally: Ella Striem-Amit, Xiaoying Wang.  Correspondence and requests for materials should be addressed to E.S-A. (email: striemamit@fas.harvard.edu) or to Y.B. (email: ybi@bnu.edu.cn)

How do we represent concepts that extend beyond our perceptual experience, concepts like "freedom" and "justice", which have no clear external referent? And how do blind people represent concepts such as rainbow, whose referent is perceptible only visually and comprised of colors, which are uniquely visual qualia?

Various studies have addressed the neural correlates of concrete and abstract concepts[1–4]. Because concrete concepts, like "cup", have perceptible features, such as shape, size and color, whereas abstract concepts, like "freedom", lack sensory features, it has been proposed that the latter type of concepts rely more heavily on semantic or verbal information[5,6]. The investigation of how abstract concepts are represented has been considered an important way to understand knowledge representation in the brain. Traditionally, this has been tested by comparing brain responses to abstract and concrete words. This comparison has revealed large-scale networks of regions associated with abstract concepts involving language and more broadly multimodal processing areas and concrete concepts involving modality-specific areas[2,7–9]. Among these areas, the left anterior temporal lobe (ATL) has been deemed to play a central role in the representation and retrieval of semantic and conceptual information[1,4,7,9–11].

However, there are additional differences between abstract and concrete concepts beyond the existence of external sensory referents. Abstract concepts tend to be learned later in life, to be less familiar[12,13], and some of them refer to social or emotional contents[14,15]. The latter factor, emotional responses associated with particular concepts, has been argued to provide an emotional (internal) "sensory" referent for some concepts[16]. The involvement of differential emotional arousal for different words may be thought to provide sensorially perceptible features for certain domains of abstract concepts, contributing to an ongoing debate about the role of sensory features in concept representation[3,17–21]. Furthermore, abstract words differ from concrete ones in their linguistic properties, in that they are more ambiguous and their interpretation depends more on context-dependent variation[22,23]. Therefore, the difference between classical abstract and concrete concepts in terms of their sensory features is confounded by additional factors. Furthermore, abstract and concrete concepts differ in an additional important dimension, beyond their sensory perceptibility: their mere referentiality. Concrete concepts generally refer to external objects or referents which can be "pointed" to in the world, whereas abstract concepts do not. Nevertheless, these two dimensions are nearly impossible to be teased apart in most circumstances as most referents are intrinsically sensible.

How can the effects of sensory perceptibility and experience, as well as that of referentiality/ objecthood, be tested then? Here we take a unique approach to overcome the various confounds listed and investigate the roles of these conceptual dimensions directly, by using a special population that does not have access to sensorially perceptible referents for otherwise concrete object concepts, thereby eliminating the confounds mentioned above. In this study we chose to focus on the effect of imperceptibility. To this aim we studied a group of people born blind as they were presented with concepts that have both object referents and sensory-accessible features ("cup"); concepts that have external referents but are perceivable through vision alone, and thus are sensorially-inaccessible referents to the congenitally blind ("rainbow"); and abstract concepts without referents or sensory features, which do not refer to emotional or social relations ("freedom"). This gradient of concepts between fully concrete object and fully abstract non-object concepts in the blind allows us to separate sensory components from those of objecthood and study their neural correlates.

## Results

**Abstract concept preference in the brain.** Two experiments (Experiment 1, block design; Experiment 2, event-related design) were conducted to inspect how abstract information, in particular those aspects relating to imperceptibility ("rainbow" vs. "rain" in congenitally blind) and objecthood ("rainbow" vs. "freedom"), is represented in the brain.

To explore the effect of these factors within the hypothesized network involved in processing abstract information, we first localized brain preference for classical abstract concepts, chosen carefully as to not arouse strong emotional responses (see Supplementary Table 2). We plotted the preferential response to abstract concepts in the data from Experiment 1. Similar to previous reports[7,8], abstract concepts ("freedom", compared to concrete every-day objects that are similarly familiar to the blind; "cup"; see Fig. 1a two right-most columns) evoked significant activation in multiple regions, mainly left-lateralized, in the combined subject group (Fig. 1b; for similar findings in each group separately and the reverse contrast see Supplementary Fig. 1). These included the inferior frontal lobe, superior temporal sulcus and anterior temporal lobe (ATL), both in the anterior superior temporal plane, as well as below it towards the temporal pole. These regions did not show a significant difference between the groups (See Supplementary Fig. 2), supporting the validity of using the blind group to study the representation of abstract concepts. A more stringent contrast, in which the abstract concepts condition was further required to also elicit significant positive activation (abstract > concrete AND abstract > baseline), limited this network to the left hemisphere, and within the ATL, mainly to the dorsal and lateral aspects (Fig. 1c).

**Imperceptibility – dorsal ATL.** We then investigated which of those regions showing preference for abstract concepts were sensitive to the absence of sensory information, as opposed to sensitivity to the existence of external referents or to other confounding factors. To do so, we examined brain activity in people blind from birth (Table 1) for concepts that have external referents in the world, but are perceptible only through the visual sensory modality, and are thus imperceptible to a congenitally blind person (e.g. "rainbow"). For the blind, these stimuli do not have sensory correlates for their defining characteristics. We compared these concepts to other concepts from the same content domain (in the case of rainbow, astral/weather phenomena) which also have external referents with sensory features in other senses, and are thus sensorially available, perceptible, to the blind (for example, "rain"; sensory perceptibility was rated by blind subjects; see methods). Imperceptible and perceptible concepts were chosen from three different content domains to avoid domain-specific effects: astral/weather phenomena (e.g. "rainbow" vs. "rain"), scenes ("island" vs. "beach") and object features (colors vs. shapes, e.g. "red" vs. "square"). Importantly, the imperceptibility comparison – ANOVA of the full design, comparing the perceptible and imperceptible concepts across domains (Fig. 1a, comparing dark red and blue across the first three left-most columns) – did not significantly differ in any of the various potentially confounding factors: general concreteness/abstractness, imageability, age of acquisition, familiarity, semantic diversity, emotional valence or arousal (mixed effects ANOVA, $F(1,58) = 0.1$, $p = 0.76$, $\eta^2 = 0.0017$, for stimuli ratings see Supplementary Table 1, for complete post hoc $t$ test results see Supplementary Table 2).

Given that the imperceptible concepts are sensorially inaccessible only to the blind, we expected regions differentially engaged due to the sensory imperceptibility of concepts to show different responses for the blind and sighted subjects in our experimental

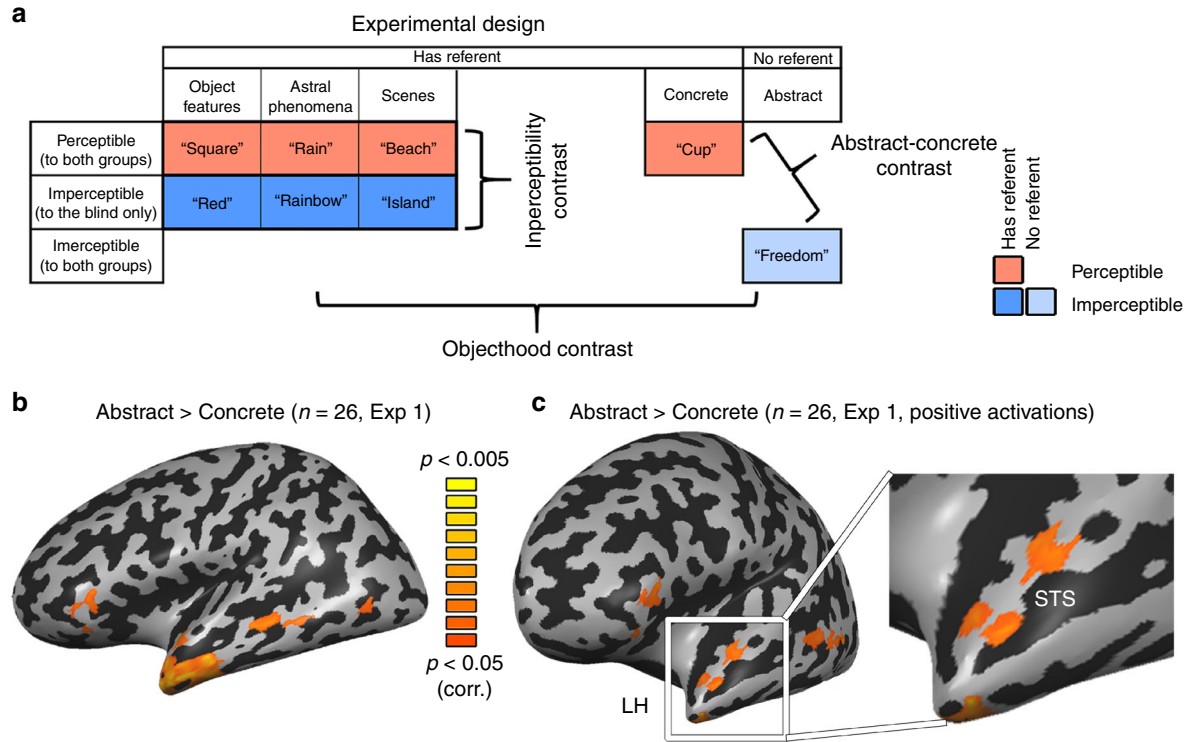

**Fig. 1** Abstract concept preference found in ATL. **a** The experimental design is depicted, along with examples of the stimuli. The row effect is that of perceptibility: items which are either at least partially perceptible to both blind and sighted (red color), imperceptible completely to the blind (blue color), or Imperceptible to both groups (light blue). The column effect is that of objecthood/referentiality. The first four columns have external referents (dark red or blue colors, depending on perceptibility), whereas the fifth one (abstract concepts; e.g. "freedom"; light color) does not. Within the first three columns, different content domains of concepts which have external referents are shown (object features, astral phenomena and scenes). For the imperceptibility contrast, all three content domains are compared between imperceptible and perceptible concepts (dark blue vs. dark red; based on the perceptual abilities of the blind). For the objecthood contrast, imperceptible concepts without referent (abstract concepts, light blue) are compared with imperceptible concepts with referents (dark blue), and particularly with the astral objects. Data was collected from two experiments: Experiment 1, a block-design, and Experiment 2, an event-related design. **b** A contrast of typical abstract words (e.g. "freedom") with concrete everyday objects (e.g., "cup") in the combined subject group ($n = 26$) shows a left-lateralized fronto-parietal network consistent with previous findings[2,7,8]. This and all other univariate statistical parametric maps originate from Experiment 1. **c** A more stringent contrast requiring the abstract concepts to also generate significantly positive activation focuses the activation to the left hemisphere, and in the ATL to its dorsal and lateral aspects (data from Exp. 1). This contrast is also presented in an anterior view, focusing on the anterior temporal lobe (ATL)

**Table 1 Blind subjects characteristics**

| Subject | Gender | Age | Years of education | Handedness | Cause of blindness | Light perception |
|---------|--------|-----|--------------------|------------|--------------------|------------------|
| B1 | M | 36 | 12 | Bi | Congenital microphthalmia | None |
| B2 | M | 22 | 15 | R | Congenital microphthalmia | None |
| B3 | M | 33 | 12 | R | Congenital microphthalmia; microcornea | None |
| B4 | M | 48 | 12 | R | Congenital glaucoma | None |
| B5 | F | 46 | 9 | R | Congenital glaucoma | None |
| B6 | M | 40 | 12 | R | Congenital leukoma | Faint |
| B7 | F | 50 | 12 | R | Cataracts; congenital eyeball dysplasia | Faint |
| B8 | M | 57 | 12 | R | Congenital eyeball dysplasia | None |
| B9 | F | 43 | 12 | R | Congenital glaucoma | None |
| B10 | M | 48 | 12 | R | Congenital microphthalmia; cataracts; leukoma | None |
| B11 | M | 63 | 9 | R | Congenital glaucoma; leukoma | None |
| B12 | F | 41 | 12 | R | Congenital optic nerve atrophy | Faint |

design. We computed an ANOVA model for a domain X imperceptibility X group effect in Experiment 1 (a block-design experiment) and looked for areas showing a group X imperceptibility interaction, different responses based on perceptibility in the two groups, across concept domains. Among the brain regions showing preference for abstract concepts in both groups, only the left ATL also showed such an interaction, in two clusters in the superior ATL (Fig. 2a; see the overlap between these contrasts in Fig. 3c, for activation profile of the other regions, such as STS and IFG, see Supplementary Fig. 3). Therefore, we focused our analyses to the ATL, long suggested to play a major role in the processing of abstract concepts[1,4,7,10,11].

To further explore the activation pattern in this region, we sampled the areas showing interaction in Experiment 1 using an

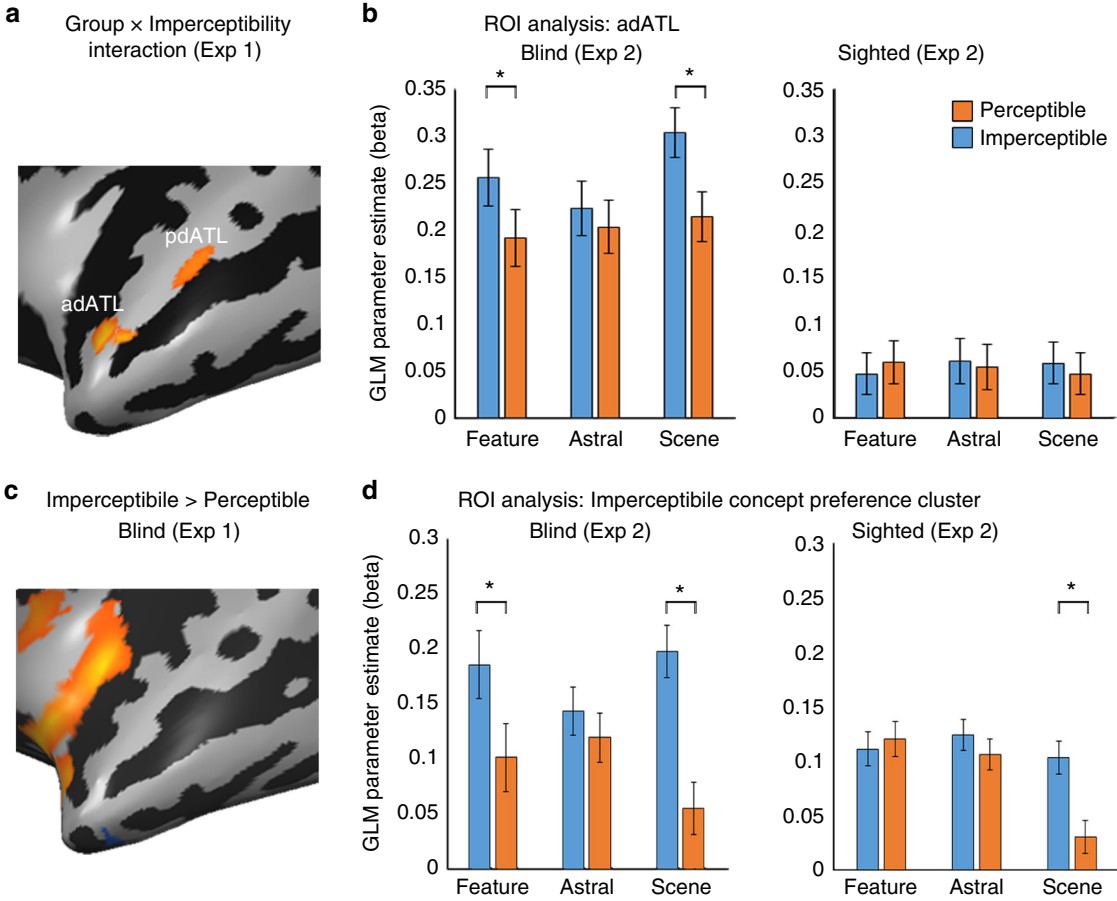

**Fig. 2** Imperceptible concepts processing is supported by the left dorsal ATL. **a** To probe for the effect of sensory feature perceptibility, we compared brain activity in people blind from birth and sighted controls, in response to concepts which have external referents in the world, but are perceptible only through the visual sensory modality, and are thus imperceptible to a blind person (e.g. "rainbow") as compared to concepts whose referents are sensorially perceptible also to the blind (through other modalities; e.g., "rain"). An area which is sensitive to imperceptibility of concepts should respond differently in the two groups for this contrast, as visually-dominant concepts are fully perceptible to the sighted subjects. The ANOVA effect of Group X Imperceptibility interaction across content domains shows two clusters in dorsal ATL which respond differently in the blind and sighted to the presented words based on their perceptibility (adATL and pdATL; data from Exp. 1). **b** The anterior cluster shown in Fig. 2A, labeled adATL, shows a preference for imperceptible concepts across concept domains (object features, astral phenomena and scenes) only in the blind group (data from independent Exp. 2). Error bars represent standard error of the difference between means for the perceptible and imperceptible words in each content domain. Asterisks represent statistically significant difference between perceptible and imperceptible concepts (paired $t$ test, t(22) > 3.505, $p < 0.05$, Bonferroni corrected for multiple comparisons). **c** Preferential activation for imperceptible vs perceptible concepts in the blind group, affects the left dorsal ATL (data from Exp. 1). **d** The dorsal ATL cluster showing preferential activation for imperceptible concepts (shown in Fig. 2C) shows a preference for imperceptible concepts across concept domains (object features, astral phenomena and scenes) only in the blind group (data from independent Exp. 2). Error bars represent standard error of the difference between means for the perceptible and imperceptible words in each content domain. In addition to the significant main imperceptibility effect in the blind, asterisks represent statistically significant difference between perceptible and imperceptible concepts (paired $t$ test, t(22) > 3.505, $p < 0.05$, Bonferroni corrected for multiple comparisons)

additional independent data set: Experiment 2, an event-related design with the same participants scanned in a separate session. Henceforward, all univariate map analyses originate from data from Experiment 1, and the bar plots of activity provide confirmatory evidence from Experiment 2.

In the anterior cluster of the interaction map, in left anterior dorsal ATL (cluster labeled adATL), we find that the interaction manifested in heightened activity in the blind group for imperceptible concepts across the three content domains (Fig. 2b, data sampled from the independent experiment 2 in adATL). For detail of the exploration of the posterior cluster (labeled pdATL in Fig. 2a), which does not show an activation pattern consistent with an overall effect of imperceptibility in the blind, see Supplementary Notes and Supplementary Figure 6. A post hoc contrast of imperceptible concepts as compared to the perceptible

counterparts in the blind in adATL (in Experiment 1 data) showed a significant effect of imperceptibility, in a slightly more dorsal part of ATL, in the anterior superior temporal plane (Fig. 2c). The same area was found also as a main effect in an imperceptibility X domain ANOVA model analysis in the blind (Supplementary Figure 6C). Specifically, when inspecting this superior ATL imperceptibility cluster in the data from Experiment 2, it showed a main effect of imperceptibility in the blind (F (1,11) = 8.63, $p < 0.05$; see sampled data in Fig. 2d), a significant preference for imperceptible concepts, but no domain effect ($p > 0.4$) or interaction ($p > 0.61$). That is, sensory perceptibility affects this region independently of content domains. Importantly, the sighted group showed no such effect in this region (all effects and interaction $p > 0.82$), and a combined ANOVA with both groups in this ROI in Experiment 2 revealed an imperceptibility X group

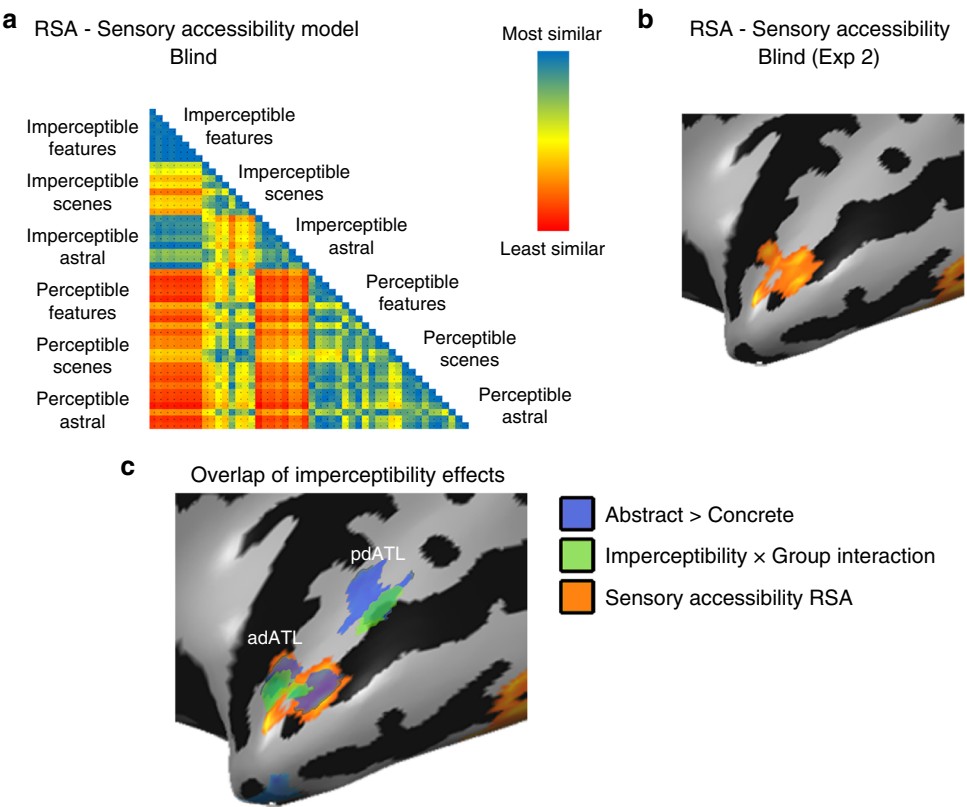

**Fig. 3** Concepts' Imperceptibility is mapped in left dorsal ATL pattern. **a**, **b** Multivariate representational similarity analysis (RSA) was computed comparing a behavioral matrix based on ratings of the blind subjects of the sensory perceptibility of the concepts (**a**; representational dissimilarity matrix) with the neural patterns in Exp. 2 in a searchlight manner across the brain. Sensory perceptibility correlation (**b**) was found in the dorsal ATL, overlapping the effects of imperceptibility X group interaction and abstract concepts preference. For additional control RSA analyses see Supplementary Fig. 4. **c** The main effects from Figs. 1–3 are shown together, to reveal the overlap in the dorsal ATL between preference for abstract concepts (over concrete ones; Fig. 1c; depicted in blue; data from Exp. 1), Imperceptibility X Group interaction (Fig. 2a; depicted in green; data from Exp. 1) and the sensory perceptibility RSA (**b**; depicted in orange; data from Exp. 2)

interaction (F(1,24) = 5.46, $p < 0.05$), supporting the absence of visual experience as the factor behind the imperceptible/perceptible category differences.

The univariate analyses reported here show a preference for imperceptible concepts in left dorsal ATL. Converging evidence from multivariate analyses further supports the role of imperceptibility in determining concept property preferences in dorsal ATL. Using behavioral ratings of the perceptible and imperceptible objects in the congenitally blind group, we computed a dissimilarity matrix of the stimuli based on their sensory perceptibility and accessibility (Fig. 3a). A multivariate comparison of the neural similarity matrices from the single-item-level event-related data (Experiment 2) in the blind, with this model of imperceptibility (searchlight representational similarity analysis; RSA), shows that the anterior dorsal ATL response pattern indeed varies based on this parameter (Fig. 3b; peak values t(11) = 5.46, $p < 0.0005$). This cluster overlaps the area showing the abstract > concrete effect as well as the group X imperceptibility interaction (Fig. 2a, adATL; see overlap in Fig. 3c). The RSA effect in dorsal ATL is found both when using ratings of sensory perceptibility produced by the blind participants scanned in this study and a group of blind participants who did not participate in the fMRI experiment ($n = 6$, Supplementary Figure 4A,B, see blind subject characteristics in Supplementary Table 3). Moreover, the "visualness" of a stimulus (as rated by an external group of sighted participants, ratings which are negatively correlated to the blind sensory perceptibility ratings; Supplementary Figure 4C,D) also correlated with the neural pattern of the activity in dorsal

ATL in the blind. Lastly, to control for any collinearity of the sensory perceptibility of the concepts with other behavioral ratings, we replicated the RSA of the sensory perceptibility of the concepts while using behavioral ratings of abstractness, imaginability, manipulability, emotional valence and emotional arousal, as well as referentiality/objecthood (which is correlated with imperceptibility; Pearson's $r^2 = 0.3$, $p < 0.001$) as nuisance regressors. A sensory perceptibility correlation was still found in the dorsal ATL, controlling for other factors which may affect abstract concept processing (Supplementary Figure 4F,G). As these ratings do not reflect the absence of sensory perceptibility for these concepts in the sighted, it is not surprising that no similar correlation between these behavioral matrices is found for the sighted neural data in dorsal ATL (t(13) < 1.53, $p > 0.15$). Therefore, evidence from both univariate and multivariate analyses support the role of left dorsal ATL in processing imperceptible concepts in the blind, suggesting that this region's response to abstract concepts is affected by the absence of sensory information regardless of objecthood/referentiality and other confounding factors.

**Objecthood and referentiality – Lateral ATL**. Is there also a preference for abstract concepts over concrete ones that can be explained by other dimensions of abstractness, such as the absence of an external referent or the absence of objecthood? To study these possibilities, we compared abstract concepts that are both physical referent-free and devoid of sensory correlates (e.g., "freedom") to concepts that have referents but no sensory

correlates (e.g., "red", "island", "rainbow", imperceptible in the blind; see Fig. 1a, comparing light and dark blue). This comparison allows us to discount the contribution of sensory information. The contrast activated the more lateral ATL regions, extending anteriorly towards the temporal pole (Fig. 4a). Since some of our imperceptible concept domains are object features ("red") and scenes ("island") rather than classical objects themselves, we further explored separately the role of objecthood. For this contrast we compared astral/weather imperceptible concepts such as "rainbow" and "moon", which are more classical figurative objects[24], to abstract concepts (Fig. 4b). This contrast replicated the preference for abstract concepts in lateral ATL, extending anteriorly towards the temporal pole. Note that these astral concepts are also comparable to abstract concepts in all relevant behavioral measures (see details in methods and Supplementary Table 2). Since the objecthood difference, independently from imperceptibility, applies to the sighted individuals as well (abstract concepts are referent-free and figurative objects have referents), we tested if the objecthood effect could be found across groups. We computed a 2-way ANOVA with objecthood and group main effects, replicating the main effect of objecthood in the lateral ATL without interaction with the group effect (Fig. 4c, see also Supplementary Figure 5 for the absence of interaction or group effect in ATL).

Lastly, although our experimental design focused on imperceptibility, and we did not have a sufficient range of item variation for this property, we carried out exploratory RSA analyses based on the behavioral ratings of referentiality/objecthood in the blind. No RSA correlation was found in ATL (Supplementary Fig 4H-J).

Overall, the lateral and anterior (pole) ATL's preference for abstract concepts over concrete ones ("freedom" over "cup"; in Fig. 1b) seems to result from a preference for external-referent-free concepts, even within imperceptible concepts. Interestingly, the effect of objecthood overlapped to some extent with areas showing the differential effect of imperceptibility between the groups, suggesting that these two dimensions are not completely orthogonal. The overlap area, the upper banks of the anterior superior temporal sulcus, appears to be affected by both factors.

**"Concrete" concepts – medial ATL**. What of the reverse effect, of having sensorially accessible properties? Again, we started by inspecting the preference in the classical contrast between concrete objects ("cup") vs. abstract concepts ("freedom") and looked within these regions for the role of perceptibility. The concrete vs. abstract contrast highlighted a known network of multisensory object processing[25,26], including the medial ATL (perirhinal cortex) in both groups (Fig. 5a; see also separately in the sighted group; Fig. 5b; Supplementary Fig. 1). If this region has a role in processing sensory features of objects, we can expect to find a preference for perceptible ("rain") over imperceptible ("rainbow") concepts in the blind, and this is indeed the case in the medial ATL (Fig. 5c; across all three content domains; comparing dark blue and red across the first three left-most columns in Fig. 1a). In analyzing Experiment 1 data we find that the medial ATL region shows a perceptibility effect in the blind (Fig. 5d) and a group X perceptibility interaction (mATL; Fig. 5e; sampled data from this ROI in Experiment 2 in Fig. 5f showed a similar non-significant trend). Therefore, the medial ATL appears to have an opposite preference than the dorsal ATL, as it prefers concepts which have perceptible sensory features. Lastly, we attempted to use RSA analysis based on the behavioral ratings of sensory perceptibility

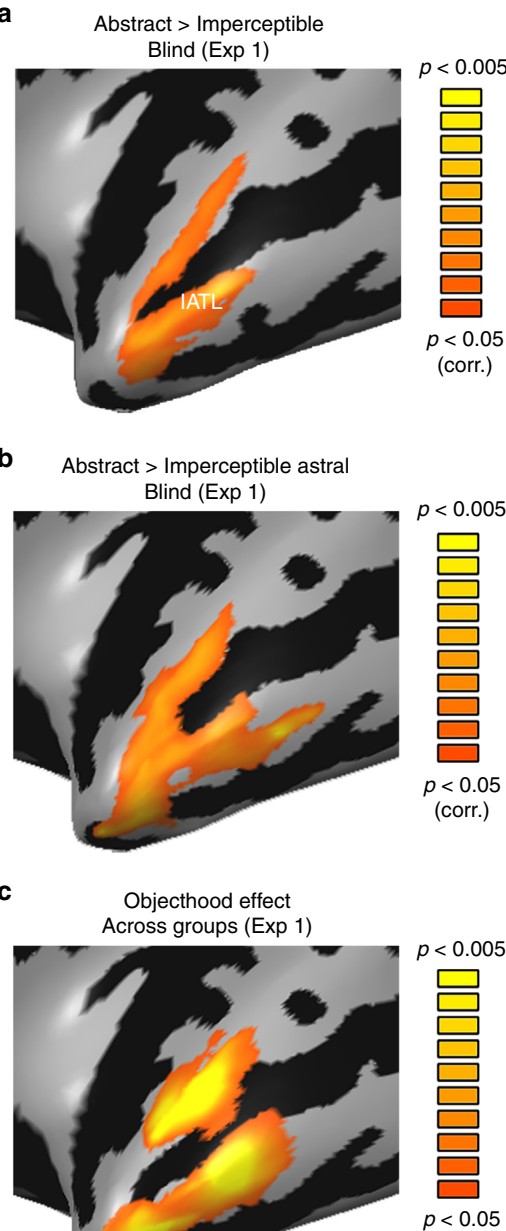

**Fig. 4** Lateral ATL shows preference for concepts without external referents. **a** The lateral ATL shows a preference for abstract, referent-free, concepts ("freedom") over imperceptible concepts (whose external referents are not sensorially accessible; "rainbow", "red" and "island") in the blind, suggesting this region's preference for abstract concepts relates to the absence of objecthood (data from Exp. 1). No areas showed significant activation for the opposite contrast, preference for imperceptible concepts over abstract ones. **b** The preference for referent-free concepts in lateral ATL is replicated as compared to the astral imperceptible concepts domain alone ("rainbow"), which are more typical (figurative) objects (data from Exp. 1). **c** The objecthood effect is found across groups when computing an Objecthood X Group ANOVA, as the difference in referentiality between abstract concepts and astral figurative objects is not limited to the blind (data from Exp. 1)

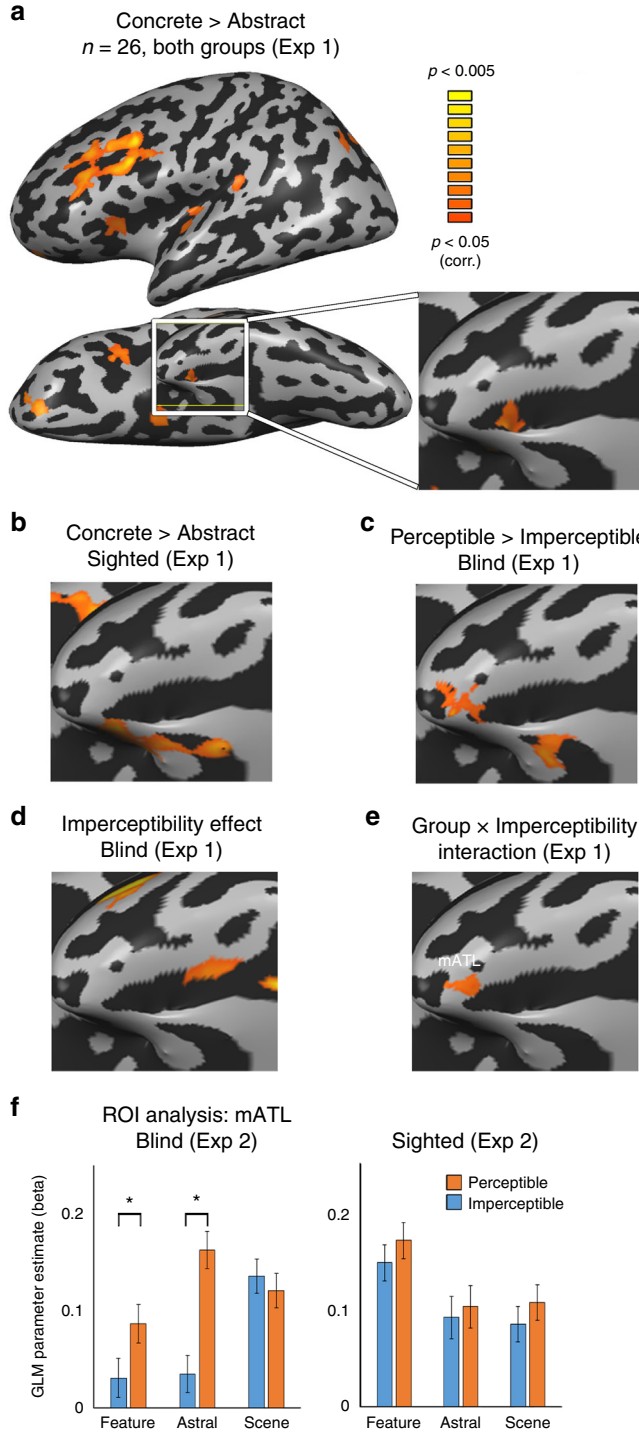

**a** Concrete > Abstract
*n* = 26, both groups (Exp 1)

*p* < 0.005

*p* < 0.05
(corr.)

**b** Concrete > Abstract
Sighted (Exp 1)

**c** Perceptible > Imperceptible
Blind (Exp 1)

**d** Imperceptibility effect
Blind (Exp 1)

**e** Group × Imperceptibility
interaction (Exp 1)

mATL

**f** ROI analysis: mATL
Blind (Exp 2)

Sighted (Exp 2)

GLM parameter estimate (beta)

Feature    Astral    Scene

Perceptible
Imperceptible

Feature    Astral    Scene

Fig. 5 Perceptible concepts processing is supported by the medial ATL. **a** A contrast of concrete everyday objects as compared with typical abstract words in the combined subject group (*n* = 26, in **a**; see **b** for sighted group separately) shows a network of regions associated with multisensory object perception (data from Exp. 1). In the ATL, the medial ATL shows preference for processing concrete objects. **b** The contrast of concrete everyday objects as compared with typical abstract words in the sighted group replicates the effect of both groups (**panel A**) in medial ATL (data from Exp. 1). **c** The contrast for perceptible vs. imperceptible concepts in the blind shows medial ATL prefers perceptible concepts (data from Exp. 1). **d** Medial ATL shows a perceptibility effect in the blind group across content domains (2-way ANOVA, perceptibility and content domain; data from Exp. 1). **e** The perceptibility effect differs between the groups in medial ATL (cluster labeled mATL), as evident from a group X imperceptibility interaction (data from Exp. 1). **f** Data sampled from mATL (the cluster shown in **e**) in the independent Experiment 2 replicates the preference for perceptible (e.g., "rain") over imperceptible ("rainbow") concepts in the blind in medial ATL, although the concepts are perceptible via non-visual modalities. Error bars represent standard error of the difference between means for the perceptible and imperceptible words in each content domain. Asterisks represent statistically significant difference between perceptible and imperceptible concepts (paired *t* test, t(22) > 3.505, *p* < 0.05, Bonferroni corrected for multiple comparisons)

by the blind and ratings of visual perceptibility by the sighted. While a small area in the medial ATL displayed RSA correlation for both analyses, this cluster was not sufficiently significant to survive the multiple comparisons correction.

**Different functional networks for aspects of ATL.** Given the different functional roles we find for different regions of ATL, we further tested if this dissociation of preferences would also manifest in having different network connectivity patterns, based on resting-state data acquired from the same participants.

We first tested the dissociation between the dorsal and lateral ATL, which appear to represent different attributes of abstract concepts (imperceptibility and non-objecthood, respectively). We computed resting-state functional connectivity (RSFC) from seeds at the peaks of the cluster showing the group X imperceptibility interaction in the dorsal ATL (adATL; Fig. 2a) and the peak of the cluster showing the abstract > imperceptible concepts in the lateral ATL (lATL; Fig. 4a) in the sighted group. Despite their difference in functional preferences, the dorsal and lateral ATL seem to belong largely to the same functional network, which includes large parts of the dorsolateral ATL and inferior frontal lobe (Fig. 6a; note the prevalence of shared RSFC marked in yellow). The spatial overlap of the activation (see detail above) and shared network suggest that these regions may be part of the same system for the processing of semantic, non-sensorially derived information. Similar connectivity patterns are found in the blind group (see Supplementary Fig. 7A,C,D), with few areas showing group differences in RSFC. This suggests that the blind brain is not differently wired in these regions, again supporting the validity of using the blind group to investigate ATL.

We investigated the potential dissociation in brain functional networks between the dorsal and medial ATL, which show contrasting roles regarding perceptibility. We plotted their joint and partial functional connectivity (RSFC; Fig. 6b) based on seeds from the peaks of the cluster showing the group X imperceptibility interaction in the dorsal ATL (adATL; shown in Fig. 2a; also used for Fig. 6a) and of the cluster showing the group X imperceptibility interaction in medial ATL (mATL; shown in Fig. 5e). The partial RSFC shows that the medial ATL is better connected to multisensory object-related regions in the frontal lobe, parietal lobe, as well as in the ventral visual cortex. This connectivity profile is consistent with the literature linking medial structures in ATL, mainly the perirhinal cortex, as the mechanism of sensory feature integration of object features[27–29]. In contrast, the dorsal ATL is more strongly connected to the lateral and anterior ATL towards the temporal pole, as well as to the inferior frontal lobe, parts of the language network. Therefore, the RSFC analysis also supports the distinct roles of these subregions of ATL. The connectivity profiles of these two

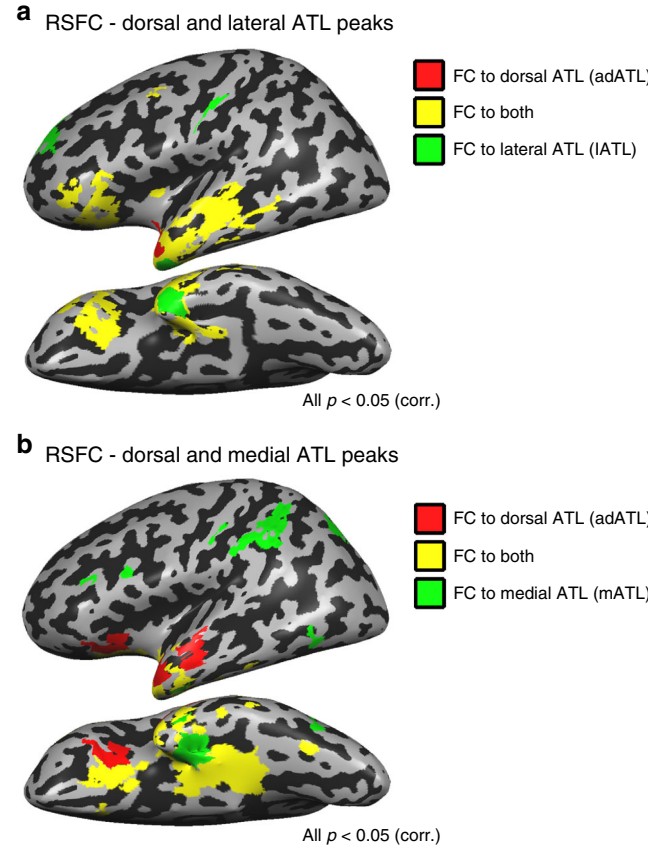

**a** RSFC - dorsal and lateral ATL peaks

- ■ (red) FC to dorsal ATL (adATL)
- ■ (yellow) FC to both
- ■ (green) FC to lateral ATL (lATL)

All *p* < 0.05 (corr.)

**b** RSFC - dorsal and medial ATL peaks

- ■ (red) FC to dorsal ATL (adATL)
- ■ (yellow) FC to both
- ■ (green) FC to medial ATL (mATL)

All *p* < 0.05 (corr.)

**Fig. 6** Different functional networks for different aspects of ATL. **a** Partial RSFC was computed from the dorsal (red; adATL) and lateral (green; lATL) ATL peaks in the sighted. Overlapping RSFC to both seeds (in yellow) is predominant, showing that these two regions belong largely the same functional network. Similar findings were evident in the blind group, and group differences in the connectivity to these seeds were minimal (see Supplementary Fig. 7A,C,D). **b** The dorsal and medial ATL regions, which show opposite preferences for perceptibility in the blind (adATL and mATL) belong to largely different functional networks in the sighted. Partial RSFC is plotted for the dorsal (red) and medial (green) ATL. Overlapping RSFC to both seeds is depicted in yellow. Similar findings were evident in the blind group, and group differences in the connectivity to these seeds were minimal (Supplementary Fig. 7B,C,E)

regions do not show large-scale differences in the blind group (Supplementary Fig. 7B,C,E).

## Discussion

We find that the response of various parts of the ATL to abstract concepts can be broken down into effects of imperceptibility and of objecthood/referentiality. Words devoid of sensorially-accessible, tangible features, either classical abstract concepts ("freedom") or words depicting visually dominant phenomena ("rainbow") in congenitally blind people, show preferred activation in the left dorsal superior ATL (Fig. 2). Supporting evidence for this are the results of the multivariate RSA which found that the activation pattern in this region correlated negatively with the level of sensory perceptibility of the concepts in the blind (Fig. 3b), even when controlling for multiple other factors, including referentiality (Supplementary Fig. 4F,G). In contrast, the lateral areas in anterior STS and the temporal pole show a preference for abstract concepts without a consistent corresponding preference for imperceptible concepts in the blind

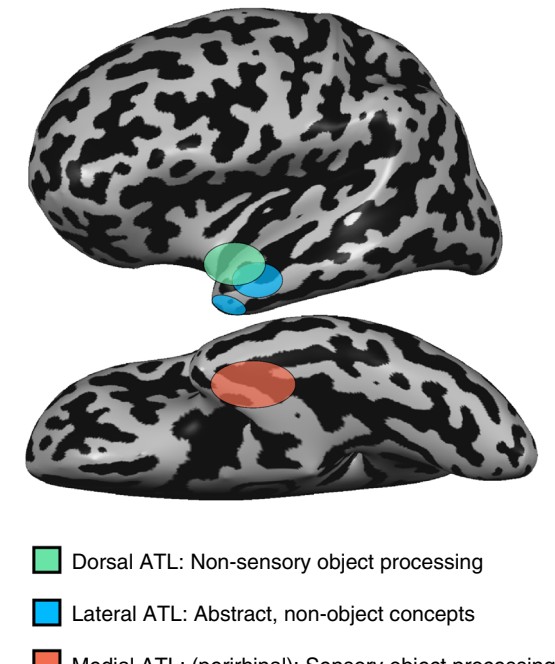

- ■ (green) Dorsal ATL: Non-sensory object processing
- ■ (blue) Lateral ATL: Abstract, non-object concepts
- ■ (red) Medial ATL: (perirhinal): Sensory object processing

**Fig. 7** Summary of ATL division-of-labor. Our results indicate that different aspects of dorsolateral ATL are engaged in the processing of abstract concepts due to a preference for imperceptible concepts (dorsal ATL, marked green) and for reference-free concepts (lateral ATL, marked blue). Medial ATL (red) shows a preference for perceptible concepts

(Supplementary Fig. 6A). Instead, their activation in response to abstract concepts exceeds that of even imperceptible concepts (Fig. 4), suggesting a role for the absence of referents altogether—the absence of objecthood—in determining this regions' representational preference. These two regions are partly overlapping and strongly functionally connected (Fig. 6a), suggesting parallel involvement in processing different but similarly amodal contents of conceptual information. Our results also further support the role of the medial ATL, the perirhinal cortex and nearby regions, in processing sensorially derived properties of concepts, as it shows a combined preference for concrete objects as well as perceptible objects in the blind (Fig. 5). This medial aspect of ATL is more strongly connected functionally to multisensory object-processing regions than the dorsal and lateral aspects of ATL (Fig. 6b). The findings reported here reveal a richly articulated neural organization of the various dimensions of the representation of abstract and concrete concepts.

First, these findings support the role of ATL in processing semantic content related to aspects of sensorially derived properties, including objecthood, while controlling for common confounds associated with the typical items used to evaluate the representation of abstract versus concrete concepts[2,12,13,16]. Multiple neuroimaging studies have emphasized the role of the superior, dorsolateral ATL, and specifically the anterior STG, in the representation and retrieval of semantic and conceptual information[1,4,7,10,11,30]. Furthermore, studies of the temporal variant of frontotemporal dementia, which results in the selective deterioration of semantic knowledge, have implicated the bilateral ATL in encoding semantic knowledge[1,10,11,31–34]. Our findings support the ATL's role in the representation of conceptual knowledge and show that the content processed in these regions can be such that it extends beyond sensory experience and object referents.

Second, this study reveals functional dissociations within the ATL—dorsal, lateral, and medial aspects of ATL—based on the

effects of perceptibility and objecthood (see Fig. 7 for illustration). This division is more fine-grained in nature within the dorso-lateral cortex. Both dorsal and lateral (middle) ATL show a preference for abstract over concrete concepts, linking them to abstract conceptual knowledge. The ATL's most dorsal aspect showed a preference for abstract concepts due to their sensory imperceptibility whereas the lateral aspects were sensitive to the absence of object referents altogether. A partial overlap between these two concept types was found in the dorsal banks of the STS, suggesting that the crucial factor is the absence of different aspects of sensory reference. Consistent with this view is the findings of functional connectivity (Fig. 6a), showing that dorsal and lateral ATL are found to belong largely to the same functional network (see also[35,36]). Therefore, it appears that both regions process abstract conceptual information, one concerning primarily imperceptible object concepts, and the other concerning concept domains which do not correspond to objects.

The distinction reported here between the roles of dorsal and lateral ATL in conceptual processing is subtle, reflecting the contribution of different aspects of abstract concepts. Much more substantial are the distinct roles of dorsolateral and ventromedial ATL in conceptual processing, reflecting the distinction between "abstract" and "concrete" concepts, respectively. This finding is in accord with research linking the medial aspect of ATL, and particularly the perirhinal cortex, to processing of sensorially derived conceptual properties of objects[1,27,37–39]. The functional dissociation we find between dorsolateral and ventromedial ATL is also in agreement with the neuropsychological literature where cases have been reported of greater deficit for concrete concepts than abstract ones in semantic dementia, some stroke patients[40–42], and in patients with ATL resection[43]—the reverse concreteness effect. Based on our results, such a phenomenon would occur in cases where temporal lobe damage involves the ventro-medial aspect of ATL, sparing (at least initially in progressive disorders) its left dorsal aspects. Evidence for medial ATL damage being associated with a deficit in processing sensorially derived conceptual properties has been demonstrated in semantic dementia patients[28]. Our results additionally provide evidence for the role of medial ATL in processing sensorially derived features of objects beyond vision and visual experience, as they revealed a preference in the blind for processing (non-visually) perceptible objects as opposed to imperceptible ones (Fig. 5). Although this region's role has been linked especially to vision and visual representations[1,44–46], we found that perceptibility, beyond the visual modality, is the critical component in activating this region. This region is distinct from the lateral and dorsal aspects of ATL and belongs to different functional networks, linking it more robustly to multisensory, object-related regions (Fig. 6b; see also ref. [36]).

Although not tested in our design, there is much evidence that object domain (e.g., animate versus inanimate[29,47–52]) and other concept properties such as their emotional/social value[16,17,53–57] play a role in the organization of conceptual processing in ATL and more posterior regions of the temporal lobe. Our findings are silent on the role of these factors, which are controlled for in our experimental design and analyses (e.g., Supplementary Fig. 4F,G). Importantly, our findings about the role of imperceptibility and objecthood in shaping the organization of abstract concepts are independent of the factors emotional response[18,24] and semantic diversity and contextual variation[22,23], which have been cited as confounding variables in investigations of abstract concepts. For non-object concepts an additional conceptual domain is that of predicates, as opposed to arguments, such as jump, plan, know, and admire. Processing these concepts involves posterior middle and superior aspects of the temporal lobe[7,58–60], reflecting a further articulation of conceptual representations across the

temporal lobes. Thus, multiple factors contribute to shaping the organization of conceptual information in ATL and the temporal lobe more generally.

The findings reported here may extend beyond the organization of the ATL and conceptual processing in the blind to reflect general principles about the factors that shape the neural organization of concepts in the sighted population. We chose to study the blind as an experimental strategy to probe into the roles of perceptibility and objecthood. There are multiple types of concepts, such as materials (e.g. colorless gases like carbon dioxide), phenomena (e.g., radiation), invisible particles (e.g., hadrons and quarks), astral remote objects (e.g., black holes), and more, which are imperceptible to us all. We did not use these types of stimuli in our design due to the difficulty in controlling for other stimulus parameters such as age of concept acquisition, word frequency, and so forth between perceptible and imperceptible concepts. Still, there is no reason to think that processing of these concept would not be supported by the same ATL regions as the imperceptible concepts to the blind. The blind showed similar activity in left ATL to traditionally abstract ("freedom") and concrete concepts ("cup", note the absence of group effects and interaction in Supplementary Fig. 2), and their connectivity in the ATL appears to be largely the same as in sighted individuals (Supplementary Fig. 7), suggesting that these brain structures do not significantly reorganize as a result of blindness, and that the findings in this group are likely to be applicable more broadly for the processing of imperceptible and non-object concepts. That said, further empirical validation would strengthen the conclusions reached here.

To summarize, the approach of studying a sensorially deprived population (the blind) has allowed us to disentangle major components of conceptual knowledge of objects and their properties: those related to perceptual properties and representations and those related to non-sensory, modality-independent information. These findings provide evidence for the neural correlates of semantic representations devoid of sensorially derived features, when controlling for multiple potential confounds, including emotional correlates. This is found across specific content domains in the blind, through both univariate and multivariate analyses, and using both dimensions of sensory perceptibility and objecthood. This amodal, sensory-independent level of concept knowledge representation is supported by the dorsolateral ATL. An additional, finer distinction reflects objecthood (e.g., "freedom" versus "rainbow" in the blind) within the larger area representing imperceptible concepts. In contrast, a preference for concrete concepts due to their sensory feature perceptibility regardless of sensory modality is supported by the medial ATL. Thus, the current findings provide important support to the neural dissociation between abstract semantic knowledge and its sensory properties.

## Methods

**Participants**. A total of 12 congenitally blind and 14 sighted subjects participated in the experiment. Participants in the blind group were between the age of 22 and 63 (mean age = 44.2 years, 8 males), and did not differ from the sighted participants in age or years of education (two-sample Welch $t$ test, df = 24, age: $p > 0.85$, years of education; $p > 0.83$). All sighted participants had normal or corrected-to-normal vision. Subjects had no history of neurological disorder. See Table 1 for detailed characteristics of the blind participants. All experimental protocols were approved by institutional review board of Department of Psychology Peking University, China, as well as by the institutional review board of Harvard University, in accordance with the Declaration of Helsinki, and all subjects gave written informed consent.

**Functional Imaging**. Images were acquired using a Siemens Prisma 3-T scanner with a 20-channel phase-array head coil at the Imaging Center for MRI Research, Peking University. The participants lay supine with their heads snugly fixed with foam pads to minimize head movement. Functional imaging data for Experiment 1

were comprised of four functional runs, each containing 251 continuous whole-brain functional volumes that were acquired with a simultaneous multi-slice (SMS) sequence supplied by Siemens: slice planes scanned along the rectal gyrus, 64 slices, phase encoding direction from posterior to anterior; 2 mm thickness; 0.2 mm gap; multi-band factor = 2; TR = 2000 ms; TE = 30 ms; FA = 90°; matrix size = 112 × 112; FOV = 224 × 224 mm; voxel size = 2 × 2 × 2 mm.

Functional imaging data for the single-item-level event-related Experiment 2 were comprised of eight functional runs, each containing 209 continuous whole-brain functional volumes using the same sequence parameters as the block-design scans. The functional scans were conducted in oblique slices to overcome some of the susceptibility artifacts affecting the ATL. Temporal signal-to-noise ratio (tSNR, the ratio of the average signal intensity to the signal standard deviation) maps were calculated and averaged across subjects for each group to assess data quality (Supplementary Fig. 8). tSNR maps show signal coverage over the anterior temporal lobes at acceptable levels for existing scan durations[61] (tSNR > 70 throughout ATL), though lowest at the temporal pole and could thus lead to lower detection power in that area. T1-weighted anatomical images were acquired using a 3D MPRAGE sequence: 192 sagittal slices; 1 mm thickness; TR = 2530 ms; TE = 2.98 ms; inversion time = 1100 ms; FA = 7°; FOV = 256 × 224 mm; voxel size = 0.5 × 0.5 × 1 mm, interpolated; matrix size = 512 × 448.

**Experimental paradigm and stimuli.** The stimuli for the experiment were spoken words, each a two-character word in Mandarin Chinese, belonging to eight concept categories (see Fig. 1a): abstract concepts (e.g., "freedom"), concrete everyday objects (e.g., "cup"), and three additional content domains, astral/weather phenomena, scenes and object features (shape and color names). Those three domains had two different categories each, one which is perceptible through non-visual senses (e.g., "rain", "beach" and "square", respectively) and the other which is perceptible only visually (e.g., "rainbow", "island" and "red", respectively), and therefore imperceptible to the blind. We used three different content domains for testing the effect of perceptibility such that domain-specific effects would be negligible. Broadly, the visually-dominant categories are those that fit the definition of "figurative", following the distinction between operative and figurative objects[24]. Operative objects, used for the perceivable categories here, are defined as those which were relatively discrete and separate from the surrounding context, and easily available to several sense modalities. Figurative elements, in contrast, are those which did not meet these criteria but nonetheless picturable and known primarily by their visual configuration. For scenes, operative, perceptible scenes were chosen such that their defining characteristics can be explored non-visually (e.g. "beach") and figurative, imperceptible, ones chosen to be too large for their overall configuration or defining features to be perceived in non-visual sensory modalities (e.g., "island"). Perceptibility ratings (the extent to which the words have associated sensory information) of the stimuli for the astral and scene categories were collected prior to the experiment by an independent group of six early-onset blind subjects without visual memory (characteristics detailed in Supplementary Table 3, subjects BS1-6) who could not participate in the fMRI study due to MR safety issues or difficulty to reach the scanning site. These subjects did not differ in education from the main sample of blind participants (two-sample Welch $t$ test, t (6) = 0.18, $p < 0.86$). Additionally, all the perceptible and imperceptible concepts were rated by the blind fMRI subjects for their sensory perceptibility several months after the scan, and were confirmed to be significantly different for all three categories (2-way ANOVA, significant perceptible-imperceptible difference across all three categories; F(1,54) = 344, $p < 0.00001$, $\eta^2 = 0.69$, post hoc Welch $t$ tests corrected for multiple comparisons for each category perceptible-imperceptible difference two-sample $t$ test t(22) > 3.32, $p < 0.005$ in all three cases).

Each category included 10 words, matched as best as possible for imageability, age of acquisition (AoA), familiarity and concreteness/abstractness, as assessed in an independent sample of 45 sighted Chinese subjects with similar levels of education (see average stimulus ratings in Supplementary Table 1). Subjects were introduced with each word separately and asked to rate it, in a scale of 1–7 for these characteristics, as reported previously[62]. Age of acquisition and familiarity are expected to be similar in the sighted and blind subjects for these concepts; even for color concepts which are a uniquely visual qualia, blind adults have shown extensive familiarity[63,64], such that they can create an approximated Newton color wheel[65], know the colors of everyday objects[66,67] (in line with a generally intact vocabulary acquisition[68]) and only a sensitive similarity measure of one specific concept category (fruit and vegetables) based on color proved to be affected by blindness[66]. Emotional valence and arousal levels were assessed in a similar manner[69] in Mandarin Chinese directly in the blind subjects, several months after the scan. Semantic diversity values were derived from previous literature[22]. Concrete objects and abstract concepts differed significantly in concreteness/abstractness, imageability and semantic diversity but not in AoA, familiarity, emotional arousal or valence (see detail for all statistical tests in Supplementary Table 2). No figurative-operative (imperceptible-perceptible) condition pairs showed significant difference in these parameters (post hoc $t$ tests, corrected for multiple comparisons; see Supplementary Table 2). Importantly, the overall imperceptible vs. perceptible design (relevant for fMRI effect depicted in Fig. 2a, c, Fig. 5c, d, e) did not significantly differ (mixed-effects ANOVA, F(1,58) = 0.1, $p = 0.76$, $\eta^2 = 0.0017$), nor did it significantly differ in post hoc contrasts in any of the behavioral parameters (post hoc Welch $t$ -tests, Bonferroni corrected, t(18) < 2.19,

$p > 0.03$, corrected $\alpha = 0.007$). The comparison of the imperceptible astral concepts to the abstract concepts (relevant for fMRI effect depicted in Fig. 4b) differed in the abstractness/ concreteness and imageability ratings but not in any other parameter (see Supplementary Table 2).

During Experiment 1, the participants kept their eyes closed and heard short lists of words in a block design paradigm (8 s blocks with 8 words each, baseline between blocks 8 s). They were instructed to detect and respond to semantic catch trials, a fruit name appearing within blocks (which occurred three times in each run; these blocks were removed from further analysis). Each run began with a 12 s rest period. Each block contained words from one of the eight concept categories.

Experiment 2 was an item-level slow event-related design and was carried out to conduct representational similarity analysis (RSA;[70]), as well as to be used as an independent data set for sampling ROI data (as the ROIs were defined from maps plotted from Experiment 1). Experiment 2 was conducted at a different scanning session on the same subjects. The stimuli were eight of the ten words from the perceptible, imperceptible and abstract categories from the main, block-design experiment stimuli. During each of the eight slow event-related runs, the subjects heard each word once, in a random order, followed by a 5 s baseline period. The subjects task was, as in Experiment 1, to detect fruit names, trials which were not further analyzed.

**Data analysis.** Data analysis was performed using the Brain Voyager QX 2.8 software package (Brain Innovation, Maastricht, Netherlands) using standard preprocessing procedures. The first two images of each scan were excluded from the analysis because of non-steady state magnetization. Functional MRI data preprocessing included head motion correction, slice scan time correction and high-pass filtering (cutoff frequency: 3 cycles/scan) using temporal smoothing in the frequency domain to remove drifts and to improve the signal to noise ratio. No data included in the study showed translational motion exceeding 2 mm in any given axis, or had spike-like motion of more than 1 mm in any direction. Functional and anatomical datasets for each subject were aligned and fit to standardized Talairach space[71]. Single subject data were spatially smoothed with a three-dimensional 6 mm full-width at half- maximum Gaussian in order to reduce inter-subject anatomical variability, and then grouped using a general linear model (GLM) in a hierarchical random effects analysis (RFX;[72]). Group analyses were conducted for the blind and sighted group separately (e.g. Supplementary Fig. 1) and for the combined blind and sighted subject group ($n = 26$, e.g., Fig. 1b, c). An ANOVA model was computed for group, stimulus domain and perceptibility, including the perceptible and imperceptible stimuli for the object features, scenes and weather/astral phenomena (Figs. 2a and 5e; comparing dark red and dark blue three left-most columns in Fig. 1a). An imperceptibility X domain ANOVA model were computed for the blind group separately to assess the direction of the interaction with the group (Supplementary Fig. 6). The minimum significance level of all results presented in this study was set to $p < 0.05$ corrected for multiple comparisons, using the spatial extent method[73] (a set-level statistical inference correction). This was done based on the Monte Carlo simulation approach, extended to 3D datasets using the threshold size plug-in for BrainVoyager QX. The correction was applied in the entire cortex for the abstract vs. concrete contrast (and vice versa; e.g., Figs. 1b and 5a) and for the group X imperceptibility inter-action, as well as for supplementary analyses in figures presented on entire cortical hemispheres. The correction was applied in the anatomically defined left ATL (the temporal lobe anterior to Heschl's gyrus) for the rest of the analyses which focused on this region. To assess the different conditions contribution to the impercept-ibility effects, we also sampled the activation GLM parameter estimates for each group and experimental condition several regions of interest. These ROIs were defined from contrasts/voxel-wise effects from Experiment 1, and sampled from the independent Experiment 2.

RSA[70] from the event-related data was computed as using CoSMoMVPA, an toolbox in MATLAB (MathWorks, Natick, MA)[74]. Dissimilarity matrices were built from behavioral ratings of the stimuli for their sensory perceptibility, as rated by the blind fMRI subjects several months after the scan (Fig. 3a). Searchlight pattern correlation analysis was computed for the unsmoothed neural patterns in the blind and sighted controls, based on the median ratings of the blind group. The mean Fisher-transformed correlation for each participant was entered into a two-tailed one-sample $t$ test against the correlation expected by chance (0) for each group. The resulting map (Fig. 3b) was corrected for multiple comparisons using the spatial extent method, as described above. A similar analysis was applied based on behavioral ratings of sensory perceptibility collected from the blind subjects who did not participate in the fMRI experiment (Supplementary Fig. 4A,B; Subjects BS1-BS6 in Supplementary Table 3), and based on behavioral ratings of visual perceptibility of the concepts as rated by an independent group of sighted participants (S3 C,D; $n = 45$). To control for any collinearity of the sensory perceptibility of the concepts with other behavioral ratings, we replicated the RSA of the sensory perceptibility of the concepts (as rated by the fMRI blind participants) while using behavioral ratings of abstractness, imaginability, manipulability, emotional valence and emotional arousal and referentiality (see below) as nuisance regressors (Supplementary Fig. 4F,G). RSA based on ratings of referentiality/objecthood collected from the blind fMRI participants and three additional congenitally blind people (total $n = 15$; see additional subject characteristics in Supplementary Table 3), while using behavioral ratings of perceptibility, abstractness, imaginability, manipulability, emotional valence and

emotional arousal as nuisance regressors. Referentiality was defined as the extent to which each concept describes something that could be pointed out in the external world (Supplementary Fig. 4H).

**Functional connectivity data analysis and MRI acquisition**. In addition to task-based data, a data set of spontaneous BOLD fluctuations for the investigation of intrinsic (rest state[75]) functional connectivity was collected while the blind and sighted subjects lay in the scanner without any external stimulation or task. Data was comprised of one functional run, containing 240 continuous whole-brain functional volumes that were acquired with the same EPI sequence and parameters as the task experiments. The first two images of each scan were excluded from the analysis because of non-steady state magnetization. After registration to individual anatomies in Talairach space, ventricles and white matter signal were sampled using a grow-region function embedded in the Brain Voyager from a seed in each individual brain. Using MATLAB ventricle and white matter time-courses were regressed out of the data and the resulting time course was filtered to the frequency band-width of 0.1–0.01 Hz. The resulting data were then imported back onto BrainVoyager for further analyses. Single subject data were spatially smoothed with a three-dimensional 6 mm half-width Gaussian. Seed regions-of-interest (ROIs) were defined from the group-level analyses of the task-data from Experiment 1. Resting state functional connectivity (RSFC) was computed from the following: (1) a cluster showing group X imperceptibility interaction in the dorsal ATL (Fig. 2a; adATL), (2) a cluster showing group X imperceptibility interaction in medial ATL (Fig. 5e; mATL), (3) a cluster in lateral anterior ATL showing preference to concepts without external referents ("freedom") over imperceptible astral ones ("rainbow"; Fig. 4a; lATL). Individual time courses from these seed ROIs were sampled from each of the sighted participants, z-normalized and used as individual predictors in group random-effect GLM analysis. Partial correlation was also computed for seeds 1 and 2 (Fig. 6b), and seeds 1 and 3 (Fig. 6a), to observe the common and separate networks to which these seeds belong. The results were corrected for multiple comparisons using the spatial extent method within the entire cortex as detailed above. Identical analysis was conducted in the blind group separately, and the joint group data was used to compute the difference in RSFC between the groups (sighted RSFC > blind RSFC, p < 0.05 corrected for multiple comparisons; see Supplementary Fig. 7).

## Data availability
The data that support the findings of this study are available from the corresponding authors upon reasonable request.

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

## Acknowledgements

We are thankful to the blind subjects who participated in our experiment, and to Hila Zadka for statistical advising. This work was supported by National Natural Science Foundation of China (31500882 to X.Y.W., 31671128 to Y.B.); Società Scienze Mente Cervello–Fondazione Cassa di Risparmio di Trento e Rovereto, by a grant from the Provincia Autonoma di Trento, and by a Harvard Provostial postdoctoral fund (to A.C.); and by the European Union's Horizon 2020 Research and Innovation Programme under Marie Sklodowska-Curie Grant Agreement 654837 and the Israel National Postdoctoral Award Program for Advancing Women in Science (to E.S-A.); National Program for Special Support of Top-notch Young Professionals (to Y.B.); the Fundamental Research Funds for the Central Universities (2017XTCX04, to Y.B.) and Interdisciplinary Research Funds of Beijing Normal University (to Y.B.).

## Author contributions

E.S-A., X.W., Y.B. and A.C. designed research; X.W. performed research; E.S-A. analyzed the data; and E.S-A., X.W., Y.B. and A.C. wrote the paper.

## Additional information

**Competing interests:** The authors declare no competing interests.

