## [Peer Review File · Nature Communications]

Reviewers' comments:

Reviewer #1 (Remarks to the Author):

Review of Striem-Amit, Wang, Bi, and Caramazza for Nature Communications: "How do blind people represent rainbows? Disentangling components of conceptual representations"

Striem-Amit and colleagues describe an fMRI experiment to distinguish neural representations for abstract and concrete concepts. Participants who were blind from birth were considered in order to examine the representation of concrete concepts that lack perceptible referents. In the fMRI scanner, blind and sighted participants listened to spoken words in Mandarin Chinese corresponding to abstract concepts, to concrete objects, or to separate sets of concepts (including astral phenomena, scenes, and object features) that could either be perceived only visually or through multiple modalities. Separate fMRI runs included (1) a block design in which participants monitored for semantic catch trials, (2) an event-related design with the same task, and (3) a resting-state scan. Based on a group-level contrast including all participants, the authors identified multiple voxel clusters in left ATL that were more responsive to abstract words than to either concrete words or to baseline. Further contrasts in blind and sighted participants, along with supplementary RSA and functional connectivity analyses, suggest functional differences between dorsal, lateral, and medial regions of left ATL.

This is a thoughtfully designed fMRI experiment with a unique population of blind participants. Descriptions of analyses and results could be improved for clarity.

MAJOR CONCERNS

1) I think that the Results may be more compelling and easier to parse if data were presented as a series of three experiments (the block-design experiment, the event-related experiment, and the resting-state experiment) rather than jumbling these results together.

For instance, Figure 1 displays ROI data alongside whole-brain contrasts. It appears (at least to a casual reader) as if these plots are showing the same data in a circular analysis. Actually, the whole-brain contrasts are from the block-design experiment, while the ROI plots are from the event-related experiment. Thus, what looks like a suspect circular analysis is actually an impressive replication.

Since the strongest evidence for functional-anatomical divisions within ATL appears to be from the whole-brain univariate analyses (from the block-design experiment), the "dorsal vs. lateral vs. medial" ATL distinctions could be made based on data from the block-design, and then replicated and further probed by analyzing data from the event-related and resting state experiments.

2) There appears to be a considerable amount of across-domain variance in the perceptibility effects that is never considered. For instance, Figure 2A is titled "No perceptibility effect in ant. STS, Blind." However, it looks like there is a significant difference between perceptible and imperceptible astral concepts (although no significance marker is included), and perhaps even a reliable cross-over interaction between astral concepts and feature concepts in this ROI.

3) The ROI analyses appear to be missing relevant statistics, as none of the ROI plots include significance markers and the error bars for these are not described. Additionally, analyses relating to representational similarity and functional connectivity were not nearly as well developed as the univariate analyses for the block-design experiment.

MINOR CONCERNS

1) Because of substantial signal dropout and artefacts near the sinuses, the anterior temporal lobes are notoriously difficult to image. I was surprised there was no discussion of measuring or correcting for image artefacts (e.g., through fieldmap correction).

2) I like the idea of considering the representation of visual concepts in blind participants, but I can additionally imagine plenty of concepts that are imperceptible everyone. For instance, there are colorless gases like carbon dioxide, there are invisible phenomena like radiation, and there are invisible particles like hadrons and quarks. How could representation of these more generally imperceptible concepts relate to the representation imperceptible concepts considered in the study?

3) Especially since the most reliable differences between concrete and abstract words were found in the medial ATL, it would be informative if Figure S1 include images of the medial surface (or the ventral surface, as in Figure 3).

4) The Methods section for the functional connectivity analyses indicates that "individual time courses from these seed ROIs were sampled from each of the sighted participants." Were functional connectivity analyses specific to just the sighted participants?

Reviewer #2 (Remarks to the Author):

In this paper, the authors addressed the role of sensory information in the neural representation of concepts across different domains. Particularly, they examined the brains of people born blind to disentangle the effects of perceptibility and objecthood. The authors started with a full abstract > concrete contrast in both blind and sighted individuals, thus to localize a network of regions responding preferentially to abstractness. They further explored this network by selecting positive activations of the abstract condition over baseline, as well as over the concrete condition, and thus obtaining a left lateralized set of regions comprising the anterior temporal lobe (ATL). Then, the authors broke down fMRI-measured responses of various parts of the left ATL into effects of imperceptibility and objecthood in the blind group. Specifically, the dorsal-superior ATL appeared to show preferred response for words devoid of sensorially-accessible, tangible features; lateral ATL for abstract, non-object concepts; and finally, medial ATL for perceptible concepts. Resting-state functional connectivity data and representational similarity analyses are integrated to support this functional subdivision of ATL, and the organization of perceptibility and objecthood as independent features represented within this region. The authors concluded that using a sensorially-deprived population allowed them to disentangle major modality-(in)dependent components of object and property conceptual knowledge.

Indeed, the current manuscript tackles with a fundamental topic of cognitive semantics, i.e. the role of sensory input and experience in forming abstract concept knowledge. In particular, the examination of people born blind (i.e., with no previous visual experience) allowed the authors to isolate the effect of sensory and referential components - perceptibility and objecthood, respectively →→ in the neural representation of abstract concepts.

The significance of this study is nonetheless undermined by some theoretical and methodological aspects, which significantly limit its conclusions.

First, the meaning of abstractness as a semantic dimension describing words and concepts is an unresolved issue, involving different theoretical positions which have not yet been sustained by sufficient psycho- and neurolinguistic experimental evidence (e.g., Borghi et al., 2014). Particularly, the Dual Coding Theory, as stated by Paivio (1971) -that the authors do cite- needs to be compared with other approaches which may justify the motivation of this study and the meaning of its results. The idea that 'classical' abstract concepts do not have clear external-sensory referents, as stated in the Abstract and the Introduction, is less accepted: Schwanenflugel and Shoben, 1983; Barsalou and Wiemer-Hastings, 2005; Xu, 2005; Kousta et al. 2011, Vigliocco et al. 2013; Dove, 2009, 2011, just to cite few works arguing against this position. Therefore, the theoretical motivation behind the first contrast is worth of a deeper discussion. While the network of regions preferring abstract concepts largely overlaps with previous results, as

stated by the authors themselves, and particularly with the 'language network', that would be no more than tangential proof that classical abstract concepts rely on non-sensory -i.e. linguistic?-information, which is the same reason why previous studies are inconclusive to this issue. And in fact, in this study a region within the language network ended up representing arguably linguistic features (objecthood and imperceptibility -ATL) while others (STS) did show inconsistent responses, suggesting that within the language network itself, activations may be explained by different factors, some linguistic, some unknown or unexplained by these specific results. This problem needs to be confronted in the discussion as well: do the authors embed their experimental choices into Paivio's theory because they find other approaches inadequate? What is their stance on grounding/embodiment of abstractness? How does the possibility of embodiment of abstract concept knowledge relate to the imperceptibility/objecthood results in the 'blind ATL' and why is the 'sighted ATL' organized differently, or in a simpler fashion?

Consequently, if congenitally blind individuals helped to isolate the sensory components in concrete vs. abstract concept distinction, I would have expected the authors to mainly determine an overlapping response for both abstract concepts in sighted and imperceptible but concrete concepts in blind people. Even though the stimuli might have been properly selected (nonetheless, the authors did not report possible biases of collinearity among variables with concreteness/abstractness that cannot be fully excluded or controlled for!), there is no convincing evidence that 'imperceptible but concrete' concepts in blind people do respond as 'truly abstract'. Equally, which would be the role of the dorsal superior ATL in sighted individuals? In the results, a 'gradient' (as the authors claim) from concrete to abstract representations is not actually reported, and these results speak, though absolutely interestingly, of the blind brain only, without stating anything on their own general validity.

In fact, imperceptibility of concrete concepts is something only pertaining to the blind group, while there is not such equivalent for the sighted, unbalancing the logic behind the contrast. Building special categories to be tested within the blind brain is undoubtedly a clever approach to the study of abstract knowledge conceptualization, but there should be one condition where "what is abstract for the blind" overlaps with "what is abstract for the sighted", and if not, the reasons are worth explaining. Is the blind brain wired differently (this may have been dealt with making a better use of connectivity analyses)? Moreover, the RSA approach, while supporting univariate results, was constructed on ratings by the blind group, and compared to the blind and sighted neural data separately, giving rise to a positive correlation with the former only. The authors justify this quite quickly, but in fact, at least for non-strictly visual concepts, sensory accessibility may have been rated by both groups, instead of just the blind one, and still be valid. As it is, it looks like slightly "double dipping", since correlation is sought between ratings given by one group and that same group's brain activity.

Second, and more importantly, which is the rationale for the authors to specifically focus on ATL? Both from a theoretical and a methodological (i.e., data masking or correction?) perspective, this is not clearly described or justified. Multiple readings, which should not be forced upon a reader, led me to the hypothesis that ATL came up from the more stringent contrast [(abstract>concrete)+(abstract>baseline)], but so apparently did the inferior frontal and middle temporal cortices. What about these areas, appearing to elicit significant positive activations together with ATL (figure 1C)? Or else, did they not survive the more stringent contrast? If not, this needs to be specified, because Figure 1C leads to think otherwise, as well as the fact that the anterior temporal cortex was further probed in the perceptibility contrast (showing mixed responses) but the inferior frontal was not. If only ATL elicited significant positive activations, figure 1C is misleading in that circling ATL is a mere figurative strategy, while maybe a wiser approach would be choosing a different color scale where positive vs. negative activations are differentiated.

Finally, the sample size for both sighted and congenitally blind individuals is relatively small for a direct univariate approach. Even if the stimuli have been previously characterized in independent samples, the size of experimental groups appears to significantly limit the reliability of the results.

Lenient statistical thresholds (0.05 corrected), the 'old-fashioned' use of inclusive regions of interest and selected univariate contrasts (e.g., ANOVA vs. 'simplified' ANOVA) appear as a compromise to deal with such small samples. In addition, in order to further confirm brain regions as isolated with univariate contrasts, it is quite unclear (and felt post hoc) why the authors arbitrarily decided to rely either on an RSA for confirming the role of sensory components, or on functional connectivity at rest for assessing the functional role of brain areas subserving objecthood. The authors could have relied all their analyses on an RSA for both sighted and blind samples.

Other addressable aspects should be properly amended in the manuscript:

- Methods: "Perceptibility ratings (the extent to which the words have associated sensory information) of the stimuli were collected prior to the experiment by an independent group of six blind subjects who could not participate in the fMRI study. Additionally, the perceptible and imperceptible concepts were rated by the blind fMRI subjects for their sensory accessibility several months after the scan, and were confirmed to be significantly different for all three categories (t-test, $p < 0.005$ in all three cases)." – more details could be provided on the six blind sample and on the statistical comparison of the sensory accessibility scores (ANOVA? Post-hoc ttests?)
- Methods: "Importantly, the overall imperceptible vs. perceptible design (relevant for fMRI effect depicted in Fig. 1D, F) did not significantly differ in any of the parameters (ANOVA, $F < 3.25$, $p > 0.08$)." – could more details on the ANOVA variables and results (e.g., df, eta-square?) be added?
- Methods: "Group analyses were conducted for the blind and sighted group separately (e.g. Fig. S1) and for the combined blind and sighted subject group ($n=25$, e.g. Fig. 1B, C)." – were not the overall group of 12 blind plus 14 sighted, i.e., 26?
- Methods: Further details and justifications on the RSA method should be provided, on the way individual behavioral data have been used for the two groups, why RSA has been used only for one condition and not for the other, etc. Moreover, it is unclear if the RSA approach is just observational or also predictive, since it seems just observational it should be specified.
- Results: if for Fig. 1F, "simplified imperceptibility X domain models were computed for the blind group separately", no sighted data should be reported.
- Results: Differences in the cortical recruitment between sighted and blind, as deducible by Figure S1, have hardly been considered. If the authors decide to proceed with univariate contrasts, a more detailed report should be provided.
- Figure 1A, the 'has referent-no referent' legend with gray and white squares is unclear to me; furthermore, the labeling with letter should be corrected in the legend. Typos in Figure 2 should also be corrected.

Reviewer #3 (Remarks to the Author):

In an elegantly designed study Striem-Amit and colleagues examined the role of sensory features in the representation of word meaning in the brain in both sighted and blind participants. They have manipulated perceptibility feature and were able to separate sensory and objecthood components of conceptual representation by examining a sensorially-deprived population. This work is novel and important because it elucidates the role of ATL in semantic representation. The research is well motivated and the methods are sound. I have a few comments.

Data acquisition: Please indicate which coil was used for data collection.

Data analyses: The description of multiple comparisons correction is unclear. The authors state that they have used GRF based on "Monte Carlo stimulation approach" ("simulation" misspelled) in BrainVoyager. It should be either GRF or simulation approach, but not both.

Sample size for each of the groups is small (12 and 14), but I note the challenges in recruiting participants from sensorially-deprived population.

Reporting results:

The number of reported participants in the methods section was 12 congenitally blind and 14 sighted individuals, for a total of 26 participants. However, the results are based on $n=25$. No description is given why data for one participant was dropped from the analysis.

In the methods section, reporting p-values as $p<0.85$ is not meaningful. I suggest to report exact p-values, or give a lower bound, as done in the results section.

I encourage the authors to make their data publically available.

Response letter to Reviewers' comments – Striem-Amit et al., “How do blind people represent rainbows? Disentangling components of conceptual representations”

Reviewers' comments:

Reviewer #1:

Striem-Amit and colleagues describe an fMRI experiment to distinguish neural representations for abstract and concrete concepts. Participants who were blind from birth were considered in order to examine the representation of concrete concepts that lack perceptible referents. In the fMRI scanner, blind and sighted participants listened to spoken words in Mandarin Chinese corresponding to abstract concepts, to concrete objects, or to separate sets of concepts (including astral phenomena, scenes, and object features) that could either be perceived only visually or through multiple modalities. Separate fMRI runs included (1) a block design in which participants monitored for semantic catch trials, (2) an event-related design with the same task, and (3) a resting-state scan. Based on a group-level contrast including all participants, the authors identified multiple voxel clusters in left ATL that were more responsive to abstract words than to either concrete words or to baseline. Further contrasts in blind and sighted participants, along with supplementary RSA and functional connectivity analyses, suggest functional differences between dorsal, lateral, and medial regions of left ATL.

This is a thoughtfully designed fMRI experiment with a unique population of blind participants. Descriptions of analyses and results could be improved for clarity.

- *We thank the reviewer for the positive and accurate evaluation. We have revised the writing according to the suggestions, and hope the revised manuscript is much clearer.*

MAJOR CONCERNS

1) I think that the Results may be more compelling and easier to parse if data were presented as a series of three experiments (the block-design experiment, the event-related experiment, and the resting-state experiment) rather than jumbling these results together.

For instance, Figure 1 displays ROI data alongside whole-brain contrasts. It appears (at least to a casual reader) as if these plots are showing the same data in a circular analysis. Actually, the whole-brain contrasts are from the block-design experiment, while the ROI plots are from the event-related experiment. Thus, what looks like a suspect circular analysis is actually an impressive replication.

Since the strongest evidence for functional-anatomical divisions within ATL appears to be from the whole-brain univariate analyses (from the block-design experiment), the “dorsal vs. lateral vs. medial” ATL distinctions could be made based on data from the block-design, and then replicated and further probed by analyzing data from the event-related and resting state experiments.

- *We thank the Reviewer for the suggestion and apologize for the unclear original presentation of the data. We have revised the manuscript according to the Reviewer's suggestion. We clarify which evidence comes from which data, describing the block-design as Exp. 1 and event-related design as Exp. 2 (scanned in two different sessions). This makes the replication aspect of the study clearly evident. All the contrast distinctions (in whole-brain univariate analyses) were conducted from the block-design experiment (exp. 1) and probed using the event-related data (exp. 2), and this is now clearly stated in the manuscript and labelled in the Figures to illustrate the replicability aspect of the*

study. We have also partly segregated the results based on the experiments and moved the functional connectivity analyses to a separate section and separate figure from the task-based findings. However, we found that because of the complexity of the presentation of multiple areas within the ATL, a full separation of the results based on experiments was cumbersome. We hope that the new organization of the manuscript makes the presentation of the results clearer.

2) There appears to be considerable amount of across-domain variance in the perceptibility effects that is never considered. For instance, Figure 2A is titled “No perceptibility effect in ant. STS, Blind.” However, it looks like there is a significant difference between perceptible and imperceptible astral concepts (although no significance marker is included), and perhaps even a reliable cross-over interaction between astral concepts and feature concepts in this ROI.

- *We thank the Reviewer for the suggestion to explore more deeply the interaction between perceptibility and domain selectivity. We originally focused on the cross-domain consistency in order to study aspects of perceptibility that are not specific to domains. We found that the blind showed a main effect of imperceptibility with no interaction with domain in the dorsal ATL (see Figure 1D, adATL). However, we agree that the findings in the posterior dorsal ATL could point to a more complex interaction between content domain and perceptibility (in the original Figure 2A; now found in Supplementary Fig. 5A). In the revised ms we report a whole-brain examination of the interaction (new Supplementary Fig. 5E). This analysis shows that content domain affects the activity in the ventral, medial ATL, in the uncus. However, only negative BOLD was found in this region for our experimental conditions, such that a clear interpretation of its significance is problematic.*

Although the whole-brain map did not reveal an interaction of domain and imperceptibility in the ant. STS area (pdATL; the areas sampled in the original Fig. 2A; this effect was not strong enough for multiple comparisons correction), we additionally analyzed the interaction in the pdATL ROI, which indeed shows a significant interaction ($F(2,1) = 5.73, p < 0.01$) in the blind group. While previously we did not conduct post-hoc t-tests for pairs of perceptible and imperceptible stimuli within each domain, upon doing so, the astral concepts pair shows a significant difference ($t = 4.86, p < p = 0.001$). However, we find it difficult to interpret this finding for only one content domain and without an a priori hypothesis. Thus, we believe future studies should investigate the interaction of imperceptibility and content domain in a targeted manner. We have expanded our discussion on the domain selectivity of parts of ATL, and the need to address these selectivities in future work. As to the absence of significance markers: given that our analyses and significance is of an interaction across domain and imperceptibility, this is not easily marked in a bar chart, and is instead detailed in the results section.

3) The ROI analyses appear to be missing relevant statistics, as none of the ROI plots include significance markers and the error bars for these are not described. Additionally, analyses relating to representational similarity and functional connectivity were not nearly as well developed as the univariate analyses for the block-design experiment.

- *We apologize for the oversight in the missing statistics. We didn't add significance markers for the bar plots as we tested significance of interaction and of main (cross-domain) effects, which are not readily marked. However, we now detail that the error bars reflect SEM, and we provide detailed statistics for all the reported analyses in the manuscript text.*

The Reviewer notes that RSA and FC analyses were not as detailed as the univariate analyses. We have corrected this imbalance in the revised version. We have added additional RSA findings related to both the dorsal ATL and the medial ATL. We replicate the original effect in dorsal ATL based on behavioral ratings of imperceptibility by independent congenitally blind raters ($N = 6$), who did not

participate in the fMRI experiment. These ratings are highly correlated to those of the fMRI participants ($r=0.89$), and, importantly, also correlate with the neural pattern of dorsal ATL (new Supplementary Fig. 3A,B). We also conducted RSA based on ratings of the “visualness” (visual dominance) of the concepts derived from an independent group of sighted raters ($n=45$), which naturally correlated negatively with the perceptibility ratings of the blind ($r=-0.67$; that is, items rated as having high visual dominance by sighted are less accessible to blinds). This also captured the relevant neural pattern variability in dorsal ATL, confirming that perceptibility is indeed the factor correlated to this region’s activity pattern, regardless of the raters (see Supplementary Fig. 3C,D). We also computed the RSA from the behavioral ratings of the original group of blind participants for sensory accessibility while controlling for multiple behavioral variables (abstractness, imaginability, manipulability, emotional valence and emotional arousal), as well as referentiality, by using them as nuisance regressors, replicating the findings of dorsal ATL (see Supplementary Fig. 3E,F,G).

In the medial ATL, we detail that applying an RSA analysis based on both blind sensory perceptibility and sighted visual perceptibility resulted in small clusters in the medial ATL, which did not survive the multiple comparisons correction. Hence, we could not generate supporting evidence from multivariate analyses for the role of medial ATL in perceptibility as we did for dorsal ATL.

We now additionally report that we attempted RSA analysis based on referentiality/objecthood of the concepts in ATL. We did not originally carry out an RSA for objecthood since our design and data in Experiment 2 (the event-related experiment) were not sufficient for strong analysis. We did not have enough stimuli to generate meaningful variability, since only one of our three perceptibility-related categories can be construed as involving classical objects and only one other relates to reference-free concepts (abstract concepts). This design was determined a priori to allow for generalization across conceptual domains in the perceptibility comparison. Nonetheless, we have attempted to compute an RSA analysis based on the existing data and newly collected behavioral ratings (Supplementary Fig. 3H) but this did not lead to significant findings in ATL. A cluster below the threshold for significance was located at lateral ATL, suggesting that a dedicated design may be able to reveal such effects in the future.

In the FC analyses, which is now a separate section from the task data, we have added functional connectivity analyses in the blind from the same seed ROIs in ATL, as well as FC group differences (new Fig S6). We detail on all these additional analyses in the revised manuscript.

MINOR CONCERNS

1) Because of substantial signal dropout and artefacts near the sinuses, the anterior temporal lobes are notoriously difficult to image. I was surprised there was no discussion of measuring or correcting for image artefacts (e.g., through fieldmap correction).

- We thank the reviewer for this comment. Details on acquisition parameters and tSNR analyses have now been added. Specifically, the scan was conducted in oblique slices to overcome some of the susceptibility artifacts affecting the ATL. An erroneous fieldmap acquisition protocol was collected during the scan, making the use of fieldmap correction impossible in this case. However, tSNR measurement of the data quality were analyzed (see new Supplementary Fig. 7) showing signal coverage over the anterior temporal lobes at acceptable levels though lowest at the temporal pole ($tSNR > 70$ throughout ATL; Murphy, Bodurka and Bandettini 2007 NeuroImage).

2) I like the idea of considering the representation of visual concepts in blind participants, but I can

additionally imagine plenty of concepts that are imperceptible everyone. For instance, there are colorless gases like carbon dioxide, there are invisible phenomena like radiation, and there are invisible particles like hadrons and quarks. How could representation of these more generally imperceptible concepts relate to the representation imperceptible concepts considered in the study?

- *We thank the Reviewer for bringing up the question (and multiple examples) of imperceptible representations in the sighted. We agree that there are multiple examples of concepts (including transparent materials, particles, mathematical and astral concepts) that are imperceptible to typically sighted people, which would likely be supported by the neural architecture of dorsolateral ATL. We have now added this topic and the examples of imperceptible concepts in the sighted in the revised manuscript.*
Specifically, we do not limit the conclusions of our findings to the blind, and believe our findings have wider implications beyond this unique population. Although we chose to study the blind as an experimental strategy to probe into the roles of perceptibility and objecthood, there are multiple types of concepts, as the Reviewer mentions, which are imperceptible to us all. The processing of these concepts is likely to be supported by the same ATL regions as the imperceptible concepts to the blind. The blind showed similar activity to the sighted in ATL to abstract and concrete concepts (note the absence of group effects and interaction in left ATL in new Supplementary Fig. 2B,C), and their connectivity in the ATL appears to be very similar to that found in the sighted (new Supplementary Fig. 6). This suggests that these brain structures do not significantly reorganize as a result of blindness, and that the findings in this group are applicable more broadly for the processing of imperceptible and non-object concepts.
We chose not to use stimuli imperceptible to the sighted due to the difficulty in controlling for other experimental parameters, such as age of concept acquisition, word frequency and so forth, between perceptible and imperceptible concepts. Furthermore it was quite difficult to find multiple (cross-domain) concept groups of this nature. These factors, added to the complexity of the design, which included 3 perceptible-imperceptible domains in the blind, and the required duration of the scan for item-based responses, led us to focus on concepts uniquely imperceptible in the blind. We agree that an interesting follow-up study would be to investigate how the sighted process concepts imperceptible to them across such various domains, and how the different domains also contribute to neural representation.
- 3) Especially since the most reliable differences between concrete and abstract words were found in the medial ATL, it would be informative if Figure S1 include images of the medial surface (or the ventral surface, as in Figure 3).
- *Added, thanks for the suggestion.*
- 4) The Methods section for the functional connectivity analyses indicates that “individual time courses from these seed ROIs were sampled from each of the sighted participants.” Were functional connectivity analyses specific to just the sighted participants?
- *In the original manuscript the FC analysis included only the sighted subjects, in order to learn about the role of ATL subparts in the typical population. We have now added a FC analysis from the same seeds of ATL also in the blind, largely replicating the same effects (see new Supplementary Fig. 6). We have additionally added a group comparison, indicating that ATL’s functional connectivity is largely similar to that found in the sighted, with little difference found in FC between the groups.*

Reviewer #2:

In this paper, the authors addressed the role of sensory information in the neural representation of

concepts across different domains. Particularly, they examined the brains of people born blind to disentangle the effects of perceptibility and objecthood. The authors started with a full abstract > concrete contrast in both blind and sighted individuals, thus to localize a network of regions responding preferentially to abstractness. They further explored this network by selecting positive activations of the abstract condition over baseline, as well as over the concrete condition, and thus obtaining a left lateralized set of regions comprising the anterior temporal lobe (ATL).

Then, the authors broke down fMRI-measured responses of various parts of the left ATL into effects of imperceptibility and objecthood in the blind group. Specifically, the dorsal-superior ATL appeared to show preferred response for words devoid of sensorially-accessible, tangible features; lateral ATL for abstract, non-object concepts; and finally, medial ATL for perceptible concepts. Resting-state functional connectivity data and representational similarity analyses are integrated to support this functional subdivision of ATL, and the organization of perceptibility and objecthood as independent features represented within this region. The authors concluded that using a sensorially-deprived population allowed them to disentangle major modality-(in)dependent components of object and property conceptual knowledge.

Indeed, the current manuscript tackles with a fundamental topic of cognitive semantics, i.e. the role of sensory input and experience in forming abstract concept knowledge. In particular, the examination of people born blind (i.e., with no previous visual experience) allowed the authors to isolate the effect of sensory and referential components - perceptibility and objecthood, respectively \neg in the neural representation of abstract concepts.

The significance of this study is nonetheless undermined by some theoretical and methodological aspects, which significantly limit its conclusions.

First, the meaning of abstractness as a semantic dimension describing words and concepts is an unresolved issue, involving different theoretical positions which have not yet been sustained by sufficient psycho- and neurolinguistic experimental evidence (e.g., Borghi et al., 2014). Particularly, the Dual Coding Theory, as stated by Paivio (1971) -that the authors do cite- needs to be compared with other approaches which may justify the motivation of this study and the meaning of its results. The idea that 'classical' abstract concepts do not have clear external-sensory referents, as stated in the Abstract and the Introduction, is less accepted: Schwanenflugel and Shoben, 1983; Barsalou and Wiemer-Hastings, 2005; Xu, 2005; Kousta et al. 2011, Vigliocco et al. 2013; Dove, 2009, 2011, just to cite few works arguing against this position.

Therefore, the theoretical motivation behind the first contrast is worth of a deeper discussion. While the network of regions preferring abstract concepts largely overlaps with previous results, as stated by the authors themselves, and particularly with the 'language network', that would be no more than tangential proof that classical abstract concepts rely on non-sensory -i.e. linguistic?- information, which is the same reason why previous studies are inconclusive to this issue. And in fact, in this study a region within the language network ended up representing arguably linguistic features (objecthood and imperceptibility - ATL) while others (STS) did show inconsistent responses, suggesting that within the language network itself, activations may be explained by different factors, some linguistic, some unknown or unexplained by these specific results.

- *We thank the Reviewer for the thoughtful comments regarding the treacherous grounds on which many of the claims regarding the nature of conceptual representations are based. We agree with the Reviewer that the abstract/concrete distinction is theoretically and empirically challenging. For this reason, we have tried to remain as close as possible to the experimental results and, while avoiding overly strong theoretical conclusion, we have tried to emphasize the importance of our results for resolution of specific claims about the organization of neural representations of concepts.*

In the revised version, we have taken greater care to further highlight the inconclusiveness of the literature on distinguishing between abstract and concrete concepts. We acknowledge this debate, as we did originally in motivating the need to control a word's internal emotional referents and contextual/linguistic variability. These considerations are now expanded to include explicit reference to this ongoing debate including works specifically mentioned by the Reviewer. The controversial nature of the concrete/abstract distinction is also the reason why we chose to 1) focus on specific dimensions (imperceptibility and referentiality); 2) in the localizer contrasts chose abstract nouns from a specific domain, "mental" concepts, rather than emotional or social relations. Our study is mute on these more controversial topics, and we intentionally chose to avoid a theoretical debate that is not empirically relevant to our current work. We now additionally acknowledge the narrower scope of our research.

In a similar manner, we have not posed the question concerning the representation of abstract concepts specifically in relation to the language network (although undoubtedly there is overlap) but rather in terms of the possible role that this network may play in supporting the neural organization of abstract concepts. Therefore, we focused our work on an area that has been implicated in abstractness and abstraction of conceptual knowledge: the ATL, rather than on the complete language network.

We realize that the Introduction in the original manuscript may have been less clear than we had hoped regarding the scope of the study. We hope that the revised version of the study more clearly describes our theoretical motivation regarding the role of the constructs perceptibility and sensory properties, and the potential role of emotional and social content in shaping the organization of different types of concepts.

This problem needs to be confronted in the discussion as well: do the authors embed their experimental choices into Paivio's theory because they find other approaches inadequate? What is their stance on grounding/embodiment of abstractness?

- *As noted above, the scope of our study was not to engage the issue of "embodiment" of abstractness. Our study is restricted to the question of how specific conceptual properties – perceptibility and objecthood (referentiality), while controlling for other variables, such as emotional features – help shape the organization of concepts in the brain. We think this an important issue in its own right even though it does not address directly the controversial issue of embodiment. Moreover, we were able to directly control for the effects of emotional valence and arousal in our partial RSA analysis, which highlights dorsal ATL while controlling for such potential internal connotations. We make it clear in the revised Introduction and Discussion that our current results are mute on the role of emotion and social constructs in the representation of abstract concepts, as well as the related explanations based on embodiment of such concepts.*

How does the possibility of embodiment of abstract concept knowledge relate to the imperceptibility/objecthood results in the 'blind ATL' and why is the 'sighted ATL' organized differently, or in a simpler fashion?

- *We do not propose that the blind subjects' ATL is differently organized as compared with the ATL of sighted people or that it processes abstract concepts in a different way. We now support this claim with data showing that the blind responses in ATL for abstract and concrete concepts do not significantly differ (Supplementary Fig. 1,2), and nor does the large-scale FC pattern of dorsal and lateral ATL, which support these concepts representations (Supplementary Fig. 6). We use the blind unique experience with specific concepts to probe for the effect of imperceptibility of concepts or their objecthood, to help reveal general principles of neural organization that also apply to the sighted. It is simply much easier to find concepts that are imperceptible to the blind, are learned at an early age,*

and have relatively high frequency which can be matched with perceptible concepts. By contrast, concepts like “gas”, “hydrogen”, “black hole” are imperceptible to sighted people as well but are typically learned later and are less frequent in language use. We have now clarified this strategy in the revised paper, and hope it is clearer.

Consequently, if congenitally blind individuals helped to isolate the sensory components in concrete vs. abstract concept distinction, I would have expected the authors to mainly determine an overlapping response for both abstract concepts in sighted and imperceptible but concrete concepts in blind people. Even though the stimuli might have been properly selected (nonetheless, the authors did not report possible biases of collinearity among variables with concreteness/abstractness that cannot be fully excluded or controlled for!), there is no convincing evidence that 'imperceptible but concrete' concepts in blind people do respond as 'truly abstract'. Equally, which would be the role of the dorsal superior ATL in sighted individuals? In the results, a 'gradient' (as the authors claim) from concrete to abstract representations is not actually reported, and these results speak, though absolutely interestingly, of the blind brain only, without stating anything on their own general validity.

In fact, imperceptibility of concrete concepts is something only pertaining to the blind group, while there is not such equivalent for the sighted, unbalancing the logic behind the contrast. Building special categories to be tested within the blind brain is undoubtedly a clever approach to the study of abstract knowledge conceptualization, but there should be one condition where "what is abstract for the blind" overlaps with "what is abstract for the sighted", and if not, the reason are worth explaining. Is the blind brain wired differently (this may have been dealt with making a better use of connectivity analyses)?

- *We agree with the Reviewer that imperceptibility of concrete concepts is more prevalent in the blind than in the sighted, which is why we chose this experimental approach. We do not claim that the imperceptible concepts are “truly abstract” to the blind; we merely state the factual claim that they cannot perceive their defining sensory features. We certainly acknowledge the difference between completely abstract and imperceptible concepts to the blind, as pointed out in our discussion of objecthood. We have added further a clarification to this effect in the revised manuscript.*

While imperceptibility is more prevalent in the blind, this does not mean there are no imperceptible concepts in the sighted. For example, particles (quark, atom), invisible gases and similar small or distant objects that do not have a typical visual representation. We chose not to use such examples in the current experiment due to the relatively late age of acquisition of such concepts, which would make it difficult to control for this difference between the imperceptible and perceptible concepts. However, we anticipate that these would also be supported by the neural architecture of the dorsal ATL. Of course, abstractness itself, encompassing both aspects (imperceptibility and absence of objecthood) pertains to fully abstract concepts which are similarly abstract to both groups and activate the dorsal and lateral ATL in both groups. We have now added a figure showing the effect of groups on the abstractness effect – the difference between abstract and concrete concepts (new Supplementary Fig. 2). Overall, the abstractness effect is found across the groups in the typical network engaged in this contrast. A group effect is limited to three small clusters: in the superior frontal cortex, right temporal pole and lateral ventral visual cortex. Therefore, sighted people certainly represent abstract concepts overlapping those represented by the blind, and these manifest in our experimental design and indeed show similar activity for both groups. In the revised manuscript, we now address the generality of our findings also in relation to the representation of abstract and imperceptible concepts in the Discussion section.

Our discussion of a gradient referred to the gradient between relatively most concrete and most abstract (devoid of perceptibility, objecthood, and emotion), when attempting to separate the effects of perceptibility and objecthood. Therefore, we meant the difference in “abstractness” between

concepts which are perceptible objects (e.g. rain) < imperceptible objects (e.g. rainbow) < imperceptible non-objects (e.g. freedom). We agree that this was not stated clearly enough, and we now detail this in the revised manuscript.

As to the wiring of the blind brain, we have added an analysis of the group differences in the connectivity patterns from the ATL regions of interest (see new Supplementary Fig. 6). These show that the FC of the ATL, including that of the aspects involved in the processing of abstract concepts due to both as well as the imperceptibility and non-objecthood parameters show largely similar connectivity in both groups. Specifically, the blind, much like the sighted, show FC from both lateral and dorsal ATL seeds in the majority of the ATL, and group differences in these seeds connectivity can be found only in few foci. Therefore, we have no reason to assume that the blind brain is significantly differently wired in these regions.

Lastly, to control for any collinearity of the sensory accessibility of the concepts with other behavioral ratings, we replicated the RSA of the sensory accessibility of the concepts while using behavioral ratings of abstractness, imaginability, manipulability, emotional valence and emotional arousal as nuisance regressors. Sensory accessibility correlation was still found in the dorsal ATL, controlling for any collinearity between sensory accessibility ratings and other factors.

Moreover, the RSA approach, while supporting univariate results, was constructed on ratings by the blind group, and compared to the blind and sighted neural data separately, giving rise to a positive correlation with the former only. The authors justify this quite quickly, but in fact, at least for non-strictly visual concepts, sensory accessibility may have been rated by both groups, instead of just the blind one, and still be valid. As it is, it looks like slightly "double dipping", since correlation is sought between ratings given by one group and that same group's brain activity.

- *RSA analysis is typically conducted by comparing behavioral ratings to brain activity for the same group (or otherwise not dissimilar – same population) of subjects, or even based on the individual ratings. In this case, to show that these ratings and the RSA analysis does not capture other potential confounding dimensions other than the effect of interest (imperceptibility), which would also be applicable to the sighted, we further computed the RSA also in the sighted as a natural control. Nonetheless, following this comment we have now also conducted an RSA analysis based on two additional behavioral measures, and present this in new Supplementary Fig. 3.*

An RSA effect in dorsal ATL is also found in the blind when using ratings of perceptual accessibility for the astral and scene stimuli, derived from an external group of blind participants, who did not participate in the fMRI experiment (N=6). These ratings were highly correlated to those of the fMRI subjects ($r=0.89$, $p<0.0001$). Additionally, an RSA effect in dorsal ATL is also found in the blind when using ratings of "visualness" of the concepts (how visually perceptible concepts are; which are negatively correlated to the blind perceptual accessibility ratings; $r=-0.67$, $p<0.0001$) derived from an external group of sighted participants ($n=45$). In both these cases no RSA effect is found in the sighted in dorsal ATL. Therefore, our RSA results do not depend on the identity of the raters.

Second, and more importantly, which is the rationale for the authors to specifically focus on ATL? Both from a theoretical and a methodological (i.e., data masking or correction?) perspective, this is not clearly described or justified. Multiple readings, which should not be forced upon a reader, led me to the hypothesis that ATL came up from the more stringent contrast [(abstract>concrete)+(abstract>baseline)], but so apparently did the inferior frontal and middle temporal cortices. What about these areas, appearing to elicit significant positive activations together with ATL (figure 1C)? Or else, did they not survive the more stringent contrast? If not, this needs to be specified, because Figure 1C leads to think otherwise, as well as the fact that the anterior temporal cortex was further probed in the perceptibility contrast (showing

mixed responses) but the inferior frontal was not. If only ATL elicited significant positive activations, figure 1C is misleading in that circling ATL is a mere figurative strategy, while maybe a wiser approach would be choosing a different color scale where positive vs. negative activations are differentiated.

- *We apologize if the description of our results was unclear as to the choice to focus on ATL. This was due to the conjunction of a vast literature focusing on this region as underlying the neural processing of abstract concepts (Binder et al., 2009; Fedorenko and Thompson-Schill, 2014; Lambon Ralph et al., 2017; Marinkovic et al., 2003; Mirman et al., 2015; Noppeney and Price, 2004; Patterson et al., 2007; Simmons and Martin, 2009), as well as our own finding that the imperceptibility X group interaction (Fig. 1D) in areas engaged in processing of abstract concepts (Fig. 1C) was found **solely** in this part of the brain. As this was the a priori defined contrast of interest in our study, we based the follow-up analyses on the areas showing these effects, and appropriately used small volume correction (SVC) within this area. Please note that supplementary figures present most of the same contrasts at a whole-cortex level, and are corrected in the entire cortex, accordingly. We now detail this motivation and accordingly the methodological approaches in the revised manuscript, as well as in the methods section.*

Finally, the sample size for both sighted and congenitally blind individuals is relatively small for a direct univariate approach. Even if the stimuli have been previously characterized in independent samples, the size of experimental groups appears to significantly limit the reliability of the results. Lenient statistical thresholds (0.05 corrected), the ‘old-fashioned’ use of inclusive regions of interest and selected univariate contrasts (e.g., ANOVA vs. ‘simplified’ ANOVA) appear as a compromise to deal with such small samples.

- *While we agree that potentially smaller effects could be revealed using a larger sample size, we feel that the converging evidence from the two experiments for the univariate analyses, as well as the convergence with the multivariate analyses provide us with reproducibly reliable results. Following another Reviewer’s comment (Rev.1) we realized that our manuscript was not sufficiently clear in emphasizing that our results include two functional experiments, a block-design experiment (Experiment 1) used for the univariate contrasts and ANOVA maps, and an event-related design experiment (Experiment 2; scanned in a different session) on the same participants, used for sampling the ROIs (defined from the block-design experiment maps) and for the RSA analysis.*

Therefore, while our maps are defined by a threshold corrected to 0.05, the sampled data from the second dataset reproduces the same effects, providing robustness to our claims. As this was not clear enough in the original submission, the revised manuscript now emphasizes which data is used for which analysis, making the replication and robustness of the data clearer.

As to the specific methodological detail, in the ANOVA designs, we have now supplemented the blind ANOVA (now found in Supplementary Fig. 5C) with the appropriate post-hoc contrasts, showing that a direct contrast between imperceptible and perceptible concepts in the blind reproduces the same effect in the blind dorsal ATL (see revised Fig. 1F-G). Please note also that the voxel-level map analyses from Experiment 1 are also reproduced by the sampled bar plots, and computed values of the ANOVA originating from experiment 2, providing reproduction for this result.

Lastly, please also note, that given the unique population studied, fully congenitally blind adults blinded due to peripheral, non-cortical etiology, it is challenging to form subject groups of the sizes typically used when scanning typically developed people. Most groups studying the blind have a similar sample size, if not smaller, and have nonetheless been able to conduct careful and stringent studies and report reliable and replicable findings (e.g. Abboud et al., Nature Communications 2015 – replicating VWFA effect from Striem-Amit 2012 Neuron; Striem-Amit and Amedi, Current Biology

2014 and Kitada et al, JNS 2014 both show EBA body part preference; van den Hurk et al., PNAS 2017 replicating similar effects in the field).

In addition, in order to further confirm brain regions as isolated with univariate contrasts, it is quite unclear (and felt post hoc) why the authors arbitrarily decided to rely either on an RSA for confirming the role of sensory components, or on functional connectivity at rest for assessing the functional role of brain areas subserving objecthood. The authors could have relied all their analyses on an RSA for both sighted and blind samples.

- *We apologize if our manuscript did not clearly state the rationale of the used analyses. The functional connectivity was used to assess the network components of all our regions of interest, in both perceptibility (Fig. 4A) and objecthood (Fig. 4B) effects, to which we now also add group comparisons (see the new Fig S6).*

In the revised manuscript we now clearly state that the RSA analysis was used to address the representational content related to perceptibility, to complement the univariate analysis. This is now presented with multiple additional controls, including controlling for collinearity with other factors. We had originally not attempted to use the same approach to target objecthood since we did not have enough stimuli to generate meaningful variability, given that only one of the three perceptibility-related categories pertains to classical objects, and only one of the categories (abstract concepts) is without referents. This was determined a priori to allow for generalization across conceptual domains in the perceptibility comparison. Please note that in presenting 8 stimuli from each of the seven categories on an item-by-item basis, in addition to the block-design experiment and resting state data already required long scans (each subject was scanned for 3 hours in two sessions for both block-design and event-related design experiments) and we could not add to it two additional categories to provide the same power to the objecthood comparison during the current experiment. Therefore, we focused the current work on an investigation of the effect of perceptibility. However, we now additionally attempted to conduct RSA of referentiality based on the existing data. We collected behavioral ratings of referentiality, defined as the extent to which each concept describes something that could be pointed out in the external world, from a blind group (which included the fMRI participants, n=15, data for the added three is found in Table S3). Behavioral ratings collected from an independent group of sighted participants was highly correlated to these ratings (R=0.92). We attempted to use RSA analysis based on the behavioral ratings of referentiality/objecthood to the blind, while using behavioral ratings of imperceptibility, abstractness, imaginability, manipulability, emotional valence and emotional arousal as nuisance regressors. No RSA correlation was found in ATL at significant thresholds (see Supplementary Fig. 3H for RDM), although a trend for an effect in lateral ATL was evident at $p < 0.05$ uncorrected. We agree that targeting objecthood with RSA in a dedicated experimental design could be an interesting idea for a follow-up experiment.

Other addressable aspects should be properly amended in the manuscript:

- Methods: “Perceptibility ratings (the extent to which the words have associated sensory information) of the stimuli were collected prior to the experiment by an independent group of six blind subjects who could not participate in the fMRI study. Additionally, the perceptible and imperceptible concepts were rated by the blind fMRI subjects for their sensory accessibility several months after the scan, and were confirmed to be significantly different for all three categories (t-test, $p < 0.005$ in all three cases).” – more details could be provided on the six blind sample and on the statistical comparison of the sensory accessibility scores (ANOVA? Post-hoc ttests?)

- *Added. The six blind subject sample was comprised of 6 early-onset blind individuals (5 congenitally blind, one blinded at age 1 year), none of whom had visual memory, and all had either no visual percept of faint light perception (but no shape or form perception). They could not participate in the fMRI study due to MR safety issues or difficulty to reach the scanning site (living in remote areas). They did not differ in education from the main sample of blind participants (t-test, $t(6) = 0.18$, $p < 0.86$). The characteristics of this group have now been added as Table S3 to the revised manuscript.*

For the statistical comparisons, we now report both the ANOVA and post-hoc contrasts. The ANOVA was a 2-way design of concept type (perceptible, imperceptible) and category (Feature, Astral phenomena and Scenes). This ANOVA yielded a significant difference between the imperceptible and perceptible concepts: $F(1,2) = 344$, $p = 4.3E-25$, $\eta^2 = 0.69$, and also yielded significant category and interaction between concept type and category. Thus we also computed post-hoc t-tests instead of ANOVAs per category for the comparison between perceptible and imperceptible concepts, corrected for multiple comparisons, where significant effects were found for all three categories ($t(22) > 3.32$, $p < 0.005$).

Thank you for directing our attention to the missing detail.

- Methods: “Importantly, the overall imperceptible vs. perceptible design (relevant for fMRI effect depicted in Fig. 1D, F) did not significantly differ in any of the parameters (ANOVA, $F < 3.25$, $p > 0.08$.” – could more details on the ANOVA variables and results (e.g., df, eta-square?) be added?

- *Added. The ANOVA was a 2-way design of concept type (perceptible, imperceptible) and behavioral measure (Concreteness/Abstractness, Familiarity, Age of acquisition, Imageability, Emotional arousal, Emotional valence, Semantic Diversity). This ANOVA did not yield a significant difference between the imperceptible and perceptible concepts: $F(1,6) = 0.06$, $p = 0.80$, $\eta^2 = 0.00004$, but did yield an interaction between concept type and behavioral measure ($F(1,6) = 2.38$, $p = 0.03$, $\eta^2 = 0.008$). Following consultation with a statistician, we now computed post-hoc t-tests instead of ANOVAs per behavioral measure, corrected for multiple comparisons, where no significant effect was found ($t(18) < 2.19$, $p > 0.03$, corrected $\alpha = 0.007$). Thanks for directing our attention to the missing detail.*

- Methods: “Group analyses were conducted for the blind and sighted group separately (e.g. Supplementary Fig. 1) and for the combined blind and sighted subject group ($n = 25$, e.g. Fig. 1B, C).” – were not the overall group of 12 blind plus 14 sighted, i.e., 26?

- *Our apologies for the mistake. The total number of subjects used was indeed 26, and this is corrected in the revised manuscript.*

- Methods: Further details and justifications on the RSA method should be provided, on the way individual behavioral data have been used for the two groups, why RSA has been used only for one condition and not for the other, etc. Moreover, it is unclear if the RSA approach is just observational or also predictive, since it seems just observational it should be specified.

- *We have added detail on the RSA to the methods section. Specifically, we now report that the RSA is observational only, and used the group average of the blind behavioral ratings to inspect the similarity in the same group as RSA is typically used, to look for similarity between behavior and neural pattern related to the dimension of perceptibility. To show that these ratings and the RSA analysis do not capture another dimension that would be processed in the same area, we further computed the RSA also in the controls. We now additionally use ratings of sensory accessibility from an independent group of blind subjects for the RSA analysis, as well as ratings of visual perceptibility*

of the stimuli by a sighted subject group, both of which show an effect in dorsal ATL in the blind and not in the sighted (see new Supplementary Fig. 3).

We did not originally compute an RSA also for objecthood since our design and data in Experiment 2 (the ER experiment) were not sufficient to do so. We did not have enough stimuli to generate meaningful variability, given only one of our three perceptibility-related categories pertains to classical objects and only one other relates to reference-free concepts (abstract concepts). This was determined a priori to allow for a generalization across conceptual domains in the perceptibility comparison. Please note that in presenting 8 stimuli from each of the seven categories on an item-by-item basis already required long scans and we could not add to it two additional categories to provide the same power to the objecthood comparison during the current experiment. We carried out an RSA analysis based on the existing data and newly collected behavioral ratings (Supplementary Fig. 3H-J) but this did not lead to significant findings in ATL. We however agree that targeting objecthood with RSA using a dedicated experimental design could be an interesting idea for a follow-up experiment.

- Results: if for Fig. 1F, “simplified imperceptibility X domain models were computed for the blind group separately”, no sighted data should be reported.

- *The simplified models reported as statistics in the manuscript were computed in the event-related data (Experiment 2), while the full ANOVA map was based on the block-design experiment (experiment 1). This was done in each group separately to provide replication and assess the direction of the interaction, showing that it originates from the blind perceptibility effect. These are now supplemented with post-hoc contrasts, showing the same results, i.e. an imperceptibility effect in the blind and not in the sighted. We have made clarifications in the Revision.*

- Results: Differences in the cortical recruitment between sighted and blind, as deducible by Figure S1, have hardly been considered. If the authors decide to proceed with univariate contrasts, a more detailed report should be provided.

- *We thank the reviewer for their suggestion to add data related to the group differences, and have revised accordingly. We now present new figures (Figs. S1, S2, S6) showing the group differences and interactions for the difference in activity for abstract vs concrete concepts, and for the functional connectivity analyses. We hope these are now exhaustive.*

- Figure 1A, the ‘has referent-no referent’ legend with gray and white squares is unclear to me; furthermore, the labeling with letter should be corrected in the legend. Typos in Figure 2 should also be corrected.

- *Corrected, we hope it is now clearer.*

Reviewer #3:

In an elegantly designed study Striem-Amit and colleagues examined the role of sensory features in the representation of word meaning in the brain in both sighted and blind participants. They have manipulated perceptibility feature and were able to separate sensory and objecthood components of conceptual representation by examining a sensorially-deprived population. This work is novel and important because it elucidates the role of ATL in semantic representation. The research is well motivated and the methods are sound. I have a few comments.

- *We thank the reviewer for the positive evaluation and helpful comments. We have revised the writing according to the suggestions, and hope the revised manuscript is clearer.*

Data acquisition: Please indicate which coil was used for data collection.

- *The coil used was a 20-channel phase-array head coil. This detail is added to the methods section.*

Data analyses: The description of multiple comparisons correction is unclear. The authors state that they have used GRF based on “Monte Carlo stimulation approach” (“simulation” misspelled) in BrainVoyager. It should be either GRF or simulation approach, but not both.

- *Our apologies for the mistake. The text was corrected omitting the reference to the GRF.*

Sample size for each of the groups is small (12 and 14), but I note the challenges in recruiting participants from sensorially-deprived population.

- *We thank the reviewer for understanding the challenge in recruiting the unique population studied, fully congenitally blind adults blinded due to peripheral, non-cortical etiology. Indeed, most groups studying the blind have a similar sample size, if not smaller, and have nonetheless been able to find significant, replicable findings (e.g. Abboud et al., Nature Communications 2015 – replicating VWFA effect from Striem-Amit 2012 Neuron; Striem-Amit and Amedi, Current Biology 2014 and Kitada et al, JNS 2014 both show EBA body part preference; van den Hurk et al., PNAS 2017 replicating similar effects in the field). We have also performed multiple experiments for further validation.*

Reporting results:

The number of reported participants in the methods section was 12 congenitally blind and 14 sighted individuals, for a total of 26 participants. However, the results are based on $n=25$. No description is given why data for one participant was dropped from the analysis.

- *Our apologies for the mistake in reporting the subject number. The total number of subjects used was indeed 26, and this is corrected in the revised manuscript.*

In the methods section, reporting p-values as $p < 0.85$ is not meaningful. I suggest to report exact p-values, or give a lower bound, as done in the results section.

- *Our apologies for the mistake. The text was intended to say, and was revised to $p > 0.85$.*

I encourage the authors to make their data publically available.

- *Thank you for this suggestion. Unfortunately, our current IRBs (in Peking University and Harvard) do not allow data sharing given that the subjects who participated in the study did not consent to their data being publicly shared (even in an anonymized way), but we have integrated a relevant statement making this possible in the consent form in our future studies.*

Reviewers' comments:

Reviewer #1 (Remarks to the Author):

My original concerns were primarily focused on the organization and statistical reporting in the Results. The authors have addressed my concerns about organization but have not addressed my concerns about statistical reporting. Remaining concerns can be easily addressed.

With respect to organization, the authors have done a good job in clarifying that the whole-brain and ROI data came from separate experiments. I am satisfied on this point.

With respect to statistical reporting, the authors clarify that error bars depict SEM and they explain that significance markers are not included because the interactions and main effects of interest are not easily marked in the displayed bar plots. Critically, however, this explanation suggests that the depicted data (as plotted) does not accurately reflect the analyses or results.

In order to make the bar plots more faithful to the reported statistics, error bars could reflect the standard error of the difference between means for Perceptible and Imperceptible referents within each domain. Significance markers above column pairs could then reflect the statistical reliability of the difference between Perceptible and Imperceptible referents within each domain. Otherwise, across-domain variance in the perceptibility effects is obscured.

Additionally, the authors now provide greater detail about statistics in the main text. However, the reporting of ANOVA results appears to be mistaken or very unconventional. For instance, on line 178 the authors report " $F(2,1)=5.73, p < 0.01$ ". Typically, the numerator of an F-statistic reflects the degrees of freedom of the model while the denominator reflects the degrees of freedom of the error. Oddly, this is not the case throughout the manuscript.

Again, these remaining concerns about statistical reporting can be easily addressed, and I commend the authors on a very strong revision.

Reviewer #2 (Remarks to the Author):

The authors properly replied to several of the criticisms that have been raised by the reviewers. The manuscript has been improved, both under the theoretical and methodological perspectives, and better highlight the elegant experimental design. The Results section still results slightly hard to follow: I recommend authors to better 'chaperon' the readers across the different steps of analyses and of results, so to be fully comprehended and appreciated in their sequentiality. Few additional elements may still be taken properly into account.

1. Even if the authors refer to the ATL as a region that plays 'a central role in the representation and retrieval of semantic and conceptual information', the experimental justification for focusing on ATL only derives from the results of 'an interaction of a contrast of GLM contrasts'... at this point, it might be useful to report (even in the Supplementary) brain responses of brain regions that emerge in Figure 1C (such as inferior frontal and mid temporal clusters), so to better appreciate the specificity of ATL behavior; indeed, the Introduction is specifically focused on ATL and, after 'an interaction of a contrast of GLM contrasts' the resulting clusters are in ATL...
2. I am not fully convinced -and the authors did not report any result in support of this - that 'the findings reported here extend beyond the organization of the ATL and conceptual processing in the blind and likely reflect general principles about the factors that shape the neural organization of concepts in the sighted population' – I suggest them to smooth their statements about this, and to clarify the speculative nature of their consideration;
3. All limitations that have been raised across the revision process by the three independent referees should be properly highlighted in the Discussion.

Reviewer #3 (Remarks to the Author):

Thank you for addressing my comments. Overall, the revised manuscript is much improved from the original.

Reviewers' comments:

Reviewer #1 (Remarks to the Author):

My original concerns were primarily focused on the organization and statistical reporting in the Results. The authors have addressed my concerns about organization but have not addressed my concerns about statistical reporting. Remaining concerns can be easily addressed.

- Thanks.

With respect to organization, the authors have done a good job in clarifying that the whole-brain and ROI data came from separate experiments. I am satisfied on this point.

With respect to statistical reporting, the authors clarify that error bars depict SEM and they explain that significance markers are not included because the interactions and main effects of interest are not easily marked in the displayed bar plots. Critically, however, this explanation suggests that the depicted data (as plotted) does not accurately reflect the analyses or results.

In order to make the bar plots more faithful to the reported statistics, error bars could reflect the standard error of the difference between means for Perceptible and Imperceptible referents within each domain. Significance markers above column pairs could then reflect the statistical reliability of the difference between Perceptible and Imperceptible referents within each domain. Otherwise, across-domain variance in the perceptibility effects is obscured.

- Thanks for this suggestion to revise the statistical reporting. We have revised the error bars to reflect the standard error of the difference between means (as opposed to the original, where they reflected the standard error of the mean of each condition separately across the subjects), and added significance markers as suggested, corrected for multiple comparisons.

Additionally, the authors now provide greater detail about statistics in the main text. However, the reporting of ANOVA results appears to be mistaken or very unconventional. For instance, on line 178 the authors report “ $F(2,1)=5.73, p < 0.01$ ”. Typically, the numerator of an F-statistic reflects the degrees of freedom of the model while the denominator reflects the degrees of freedom of the error. Oddly, this is not the case throughout the manuscript.

- Thanks for the correction of the ANOVA degrees of freedom statistical reporting, which was indeed inaccurate. These are now corrected.

Again, these remaining concerns about statistical reporting can be easily addressed, and I commend the authors on a very strong revision.

- Thank you.

Reviewer #2 (Remarks to the Author):

The authors properly replied to several of the criticisms that have been raised by the reviewers. The manuscript has been improved, both under the theoretical and methodological perspectives, and better highlight the elegant experimental design. The Results section still results slightly hard to follow: I recommend authors to better 'chaperon' the readers across the different steps of analyses and of results, so to be fully comprehended and appreciated in their sequentiality.

- We have added additional detail on the rationale of the various analyses order to further streamline the results section, we hope the revised manuscript is now easier to follow.

Few additional elements may still be taken properly into account.

1. Even if the authors refer to the ATL as a region that plays 'a central role in the representation and retrieval of semantic and conceptual information', the experimental justification for focusing on ATL only derives from the results of 'an interaction of a contrast of GLM contrasts'... at this point, it might be useful to report (even in the Supplementary) brain responses of brain regions that emerge in Figure 1C (such as inferior frontal and mid temporal clusters), so to better appreciate the specificity of ATL behavior; indeed, the Introduction is specifically focused on ATL and, after 'an interaction of a contrast of GLM contrasts' the resulting clusters are in ATL...

- We thank the reviewer for this suggestion to explicitly show the specificity of ATL activation profile. We have now added a supplementary figure (Supplementary Fig. 3) as suggested, showing the activation profile in terms of the perceptibility effect of each of the clusters from the contrast between abstract vs. concrete concepts (Figure 1C) which were not reported in the main text. These include the IFG, MTG and STS clusters, neither of which showed a significant group X imperceptibility interaction (for all clusters $F(1,22) < 1.4$, $p > 0.25$).

2. I am not fully convinced -and the authors did not report any result in support of this - that 'the findings reported here extend beyond the organization of the ATL and conceptual processing in the blind and likely reflect general principles about the factors that shape the neural organization of concepts in the sighted population' – I suggest them to smooth their statements about this, and to clarify the speculative nature of their consideration;

- We have moderated our claim for the generalizability of the findings and explicitly stated that our theoretical reasoning warrants further empirical validations.

3. All limitations that have been raised across the revision process by the three independent referees should be properly highlighted in the Discussion.

- We have addressed all the referees' comments to the best of our understanding. A highlighted version of the changes made during the revision is enclosed in the submission (for both this and the previous revision), and we hope the referees have access to it.

Reviewer #3 (Remarks to the Author):

Thank you for addressing my comments. Overall, the revised manuscript is much improved from the original.

- Thank you.

REVIEWERS' COMMENTS:

Reviewer #1 (Remarks to the Author):

The authors have satisfactorily addressed all of my concerns.

Reviewer #2 (Remarks to the Author):

The comments have been properly addressed and the novelty of the experimental question better highlighted, as both the experimental sample (e.g., blind individuals) and the study design (e.g., concepts whose referents are imperceptible to blind) represent the strengths of this study. The statements in the Discussion has been properly smoothed as the current analytical pipeline still do not support any specific causal or direct conclusion between stimuli and statistical interactions. Overall, the revised manuscript is much improved from the original.